# PATCH: LEARNABLE TILE-LEVEL HYBRID SPARSITY FOR LLMS

## ABSTRACT

Large language models (LLMs) deliver impressive performance but incur prohibitive memory and compute costs at deployment. Model pruning is an effective way to reduce these overheads, yet existing approaches face challenges: unstructured sparsity, where nonzeros can appear anywhere, preserves accuracy but yields irregular access patterns that prevent GPU acceleration, while semi-structured 2:4 sparsity is hardware-friendly but enforces a rigid 50% pattern that degrades model quality. To bridge this gap, we introduce PATCH, a hybrid sparsity framework that enables a continuous sparsity ratio between 0% and 50%. PATCH partitions weight matrices into tiles, assigning each tile to be either dense or 2:4 sparse via a learnable mask selection mechanism. This design provides fine-grained control over accuracy–acceleration tradeoffs and supports non-uniform sparsity across layers, leading to superior overall quality. Across models from 0.5B to 8B parameters, PATCH consistently narrows the gap to dense accuracy while delivering practical speedups. For instance, on LLaMA-2 7B with an A6000 GPU, PATCH achieves 1.18×–1.38× end-to-end speedup over dense baselines while improving accuracy by 0.37%–2.96% compared to the state-of-the-art 2:4 pruning method, MaskLLM. [1]

## 1 INTRODUCTION

Recent advancements in large language models (LLMs) have revolutionized natural language processing, enabling breakthroughs in understanding and generating human language (Comanici et al., 2025; Meta, 2025). These models power diverse applications, such as conversational agents and automated content creation (Suzgun et al., 2022; Zhou et al., 2023). However, their extensive parameter counts—often in the billions—result in significant memory overhead and high inference costs (Guo et al., 2024; Ma et al., 2024). This computational burden has driven the need for efficient model compression techniques.

Two primary approaches to model compression are quantization and sparsity. Quantization reduces the precision of model parameters, compressing LLMs effectively while preserving performance (Ashkboos et al., 2024; Tseng et al., 2024; Zhang et al., 2024; Saha et al., 2024). In contrast, sparsity aims to lower memory and computational demands by setting many parameters to zero (Hassibi et al., 1993; LeCun et al., 1989). However, sparsity alone struggles to maintain model accuracy while delivering practical speedups, a limitation that current research seeks to overcome.

Unstructured sparsity, which permits non-zero elements to appear anywhere in the matrix, can match dense model accuracy due to its flexibility in sparsity allocation (Sun et al., 2023; Frantar & Alistarh, 2023; Agarwalla et al., 2024). However, its irregular memory access patterns hinder acceleration on modern hardware like GPUs (Xia et al., 2023; Fan et al., 2025). As a result, unstructured sparsity fails to deliver practical speedups, motivating the search for more hardware-friendly sparsity techniques.

Semi-structured sparsity patterns, such as the 2:4 pattern (Mishra et al., 2021) supported by NVIDIA and AMD GPUs, provide practical speedups in large-scale model inference. However, unlike unstructured sparsity, which offers greater flexibility, 2:4 enforces rigid rules by requiring at least two of every four consecutive elements to be zero. This rigidity often leads to significant accuracy loss

---

[1]Code and data for PATCH are available at https://anonymous.4open.science/r/PATCH-ICLR2026

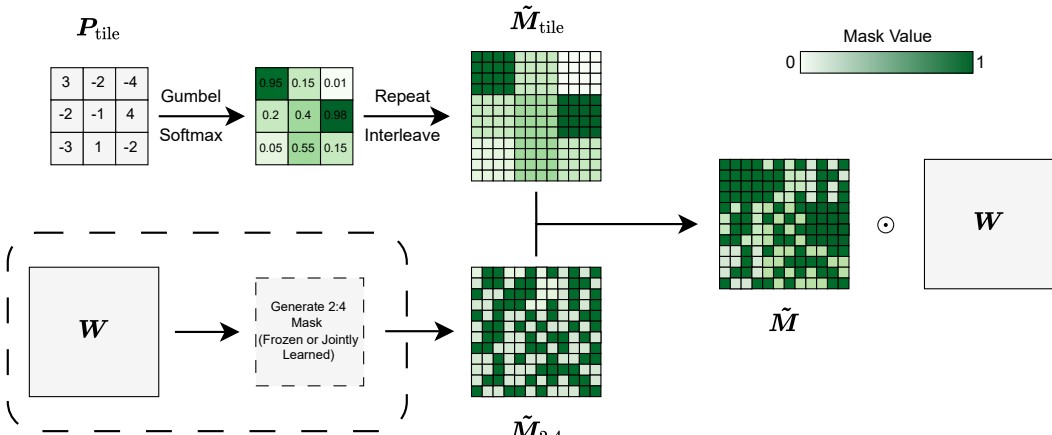

Figure 1: Illustration of the PATCH learning process for generating tile-level hybrid masks. Each tile is parameterized by a learnable distribution and sampled with Gumbel Softmax to produce $\tilde{M}_{\text{tile}}$. The dense probability is expanded and merged with a 2:4 mask $\tilde{M}_{2:4}$, which can be fixed or jointly learned during training, yielding $\tilde{M}$. The final mask assigns each tile to remain dense or follow the 2:4 pattern, enabling flexible sparsity across the weight matrix.

when models are pruned using one-shot methods (Sun et al., 2023; Frantar & Alistarh, 2023; Ilin & Richtarik, 2025; Liu et al., 2025). MaskLLM (Fang et al., 2024) mitigates this issue by learning sparsity masks end-to-end, but pruned models still lag behind their dense counterparts in accuracy. Moreover, recent studies show that sparsity should be allocated non-uniformly (adaptively) across layers for optimal performance (Yin et al., 2025; Wang & Tu, 2020; Lee et al., 2021), whereas 2:4 sparsity enforces a fixed, uniform allocation. These limitations indicate that relying solely on 2:4 sparsity is insufficient, underscoring the need for hybrid approaches.

To address the challenges of LLM pruning, while providing accelerated inference, we propose **P**runing with a Le**a**rnable **T**ile-level **C**onfiguration for **H**ybrid Sparsity (**PATCH**). PATCH learns a hybrid mask that partitions each weight matrix into hardware-friendly tiles, designating each tile as either dense (0% sparsity) or 2:4 sparse (50% sparsity). This adaptive mask allows the matrix to realize an effective global sparsity ratio anywhere between 0% and 50%, balancing accuracy in critical regions with hardware-friendly sparsity elsewhere. This design unites the hardware acceleration benefits of 2:4 sparsity with the flexibility of unstructured allocation, allowing sparsity to adapt to the varying importance of different layers. By jointly optimizing the sparsity within 2:4 tiles and the tile-level patterns during training, PATCH achieves higher accuracy than uniform sparsity across layers. Moreover, for resource-constrained settings, we offer a variant of PATCH that tunes only the dense tiles while freezing the initial 2:4 mask. Importantly, PATCH is compatible with tile-level sparsity acceleration libraries and compilers such as STOICC (Rafii et al., 2025), making it the first hybrid sparsity method to demonstrate practical speedups. For example, on LLaMA-2 7B running on a consumer-grade A6000 GPU, PATCH achieves $1.18\times$–$1.38\times$ end-to-end speedup over the dense baseline while improving accuracy by 0.37%–2.96% compared to the state-of-the-art 2:4 pruning method, MaskLLM.

## 2 PRELIMINARIES

**Differentiable Sampling.** Sampling from a categorical distribution is inherently non-differentiable, which poses challenges for gradient-based optimization. The Gumbel Softmax (Jang et al., 2016) addresses this by combining the Gumbel-Max reparameterization trick together with a softmax relaxation. The reparameterization expresses the sampling process by decoupling the deterministic log-probabilities $p \in \mathbb{R}^n$ from the stochastic perturbations $z \in \mathbb{R}^n$ introduced by Gumbel noise, which emulate random draws from the distribution. The subsequent softmax yields a differentiable approximation to categorical sampling:

$$\text{GS}(p; \tau)_k = \frac{\exp((p_k + z_k)/\tau)}{\sum_j \exp((p_j + z_j)/\tau)} \tag{1}$$

where $z_k = -\log(-\log(u_k))$ with $u_k \sim \text{Uniform}(0, 1)$. The resulting vector $\text{GS}(p; \tau) \in \mathbb{R}^n$ is a soft index vector whose entries $\text{GS}(p; \tau)_k$ represent the relaxed probability of selecting class $k$.

Additionally, the temperature parameter $\tau$ controls the hardness of the sampled index. Lower values of $\tau$ yield a more peaked distribution, causing $\text{GS}(p)$ to converge to a one-hot vector as $\tau \to 0$.

**Learnable 2:4 Mask.** MaskLLM (Fang et al., 2024) formulates 2:4 mask selection as a learnable probabilistic process over the six possible patterns. The underlying weights remain fixed, while training shifts the categorical distribution to favor masks that preserve better pruning performance. The mask for each four consecutive elements can be parameterized with a vector $p \in \mathbb{R}^{6 \times 1}$. Scaling this vector to a weight matrix $\boldsymbol{W} \in \mathbb{R}^{d_1 \times d_2}$ will result in $\boldsymbol{P}_{2:4} \in \mathbb{R}^{6 \times \frac{d_1 d_2}{4}}$ as the mask search parameters. The resulting mask can be computed as in Equation 2, where $\tilde{\boldsymbol{M}}_{2:4} \in [0, 1]^{d_1 \times d_2}$ denotes the 2:4 soft mask, obtained as a weighted average over the candidate masks, and $\boldsymbol{S} \in \mathbb{R}^{6 \times 4}$ is the matrix containing these six candidates as its rows.[2]

$$\tilde{\boldsymbol{M}}_{2:4} = \texttt{reshape}(\text{GS}(\boldsymbol{P}_{2:4}; \tau, \kappa) \times \boldsymbol{S}, \mathbb{R}^{d_1 \times d_2}) \tag{2}$$

A scaling factor $\kappa$ is also introduced in Equation 1, where it multiplies the logits $p$ before adding the Gumbel noise $z$, thereby controlling their relative influence. Small $\kappa$ values let the noise dominate, encouraging exploration across candidate masks, while larger $\kappa$ values amplify the logits and make the sampling more deterministic.

# 3 PATCH

To overcome the rigidity of fixed 50% 2:4 sparsity, we introduce PATCH. PATCH learns a structured mask—optimized on top of frozen weights—that is partitioned into tiles, where each tile decides whether its corresponding weights remain dense or are pruned with a 2:4 pattern. This design preserves accuracy in sensitive regions while exploiting hardware-accelerated sparsity elsewhere. Unlike fixed 2:4 sparsity, which enforces the same pattern across all weights, PATCH adapts at the tile level by assigning dense tiles to critical regions and sparse tiles elsewhere.

Finding the optimal allocation of dense tiles (value 1) and sparse tiles (2:4 pattern) within a mask is a combinatorially difficult problem, as the number of possible configurations grows rapidly with the number of tiles across the LLM. By also modelling this problem as a probabilistic sampling process, and adjusting the probability of each tile (and the 2:4 patterns within sparse tiles), PATCH can efficiently explore the space of configurations and converge toward masks that balance accuracy and sparsity. The mask distributions are learned end-to-end by training the Gumbel–Softmax logits while keeping the model weights frozen. We address this challenge by formulating mask selection as two coupled subproblems: *(1) selecting which tiles are dense or sparse*, and *(2) choosing the 2:4 sparsity pattern within sparse tiles*.

**Tile-based pruning of LLMs.** We associate each parameter matrix $\boldsymbol{W} \in \mathbb{R}^{d_1 \times d_2}$ with a grid of tile-level distributions, each parameterized by a learnable logit. Collectively, these form $\boldsymbol{P}_{\text{tile}} \in \mathbb{R}^{\frac{d_1}{b_1} \times \frac{d_2}{b_2}}$, where each entry specifies the unnormalized score of keeping the corresponding $b_1 \times b_2$ tile fully dense. To create a two-class distribution (keep dense vs. prune), we concatenate a fixed zero to each logit, yielding $[\boldsymbol{P}_{\text{tile}}, 0] \in \mathbb{R}^{\frac{d_1}{b_1} \times \frac{d_2}{b_2} \times 2}$. After applying Gumbel–Softmax, we broadcast the dense probabilities across their respective $b_1 \times b_2$ region (since the weighted average of the two outcomes reduces to $p_{\text{dense}} \cdot 1 + p_{\text{prune}} \cdot 0 = p_{\text{dense}}$), so that all elements of a tile receive the same mask value. Formally,

$$\tilde{\boldsymbol{M}}_{\text{tile}} = \text{GS}([\boldsymbol{P}_{\text{tile}}, 0]; \tau, \kappa)_{:,:,0} \otimes \mathbf{1}. \tag{3}$$

---

[2]We will refer to a mask value of 1 as *keeping* the corresponding weight and a value of 0 as *pruning* it.

This yields the tile-level mask $\tilde{M}_{\text{tile}} \in [0, 1]^{d_1 \times d_2}$ in Equation 3, where $\mathbf{1} \in \mathbb{R}^{b_1 \times b_2}$ is an all-ones matrix and $\otimes$ denotes the Kronecker product.

**Joint optimization with sparse mask.** To fully determine the effective sparsity pattern, the tile-level mask must be combined with the fine-grained 2:4 mask. Assuming that the 2:4 mask $\tilde{M}_{2:4}$ is generated using Equation 2, PATCH combines it with the tile mask $\tilde{M}_{\text{tile}}$ as shown in Equation 4. The resulting soft mask interpolates between dense and sparse behavior: values of $\tilde{M}_{\text{tile}}$ close to one make the tile predominantly dense, while values close to zero shift the tile toward the soft 2:4 mask pattern defined by $\tilde{M}_{2:4}$. Thus, $\tilde{M}$ can be understood as a per-tile weighted average of the dense option and the 2:4 patterns, with $\tilde{M}_{\text{tile}}$ determining the relative contribution of each. An overview of the process is provided in Figure 1.

$$\tilde{M} = \tilde{M}_{\text{tile}} + \left(1 - \tilde{M}_{\text{tile}}\right) \odot \tilde{M}_{2:4} \tag{4}$$

**Learning masks with targeted sparsity.** PATCH uses a novel regularization term to achieve a flexible 0%–50% sparsity ratio across the model by controlling the number of dense tiles. Unlike traditional regularization methods like weight decay, which produce non-deterministic sparsity ratios, our term penalizes deviations from the target sparsity, enabling precise control. This global sparsity approach prunes sensitive linear layers less aggressively while setting redundant weight elements to zero, offering greater flexibility than fixed per-layer sparsity. We directly compare global versus per-layer sparsity regularization in § 5.

**Training objective.** The overall training objective, as shown in Equation 5, of PATCH combines three components: the standard modeling loss, a sparsity regularization term that enforces the target density of the model $\rho$, and a weight regularization term (as in MaskLLM) that promotes larger weight magnitudes and gradient propagation. Formally,

$$\mathcal{L} = \mathcal{L}_{LM}\left(x; \tilde{M}_i \odot W_i\right) + \lambda_1 \left\|\frac{\sum_i \tilde{M}_i}{\sum_i \|W_i\|_0} - \rho\right\|_1 - \lambda_2 \frac{\sum_i \|\tilde{M}_i \odot W_i\|_2^2}{\sum_i \|W_i\|_2^2} \tag{5}$$

Following MaskLLM, we progressively *decrease* $\tau$ and *increase* $\kappa$ during training so that the Gumbel-Softmax distribution converges to a clear one-hot choice of mask by the end of training.

**Inference.** After training, the sign of each logit in $P_{\text{tile}}$ determines the final mask. Since a zero logit is concatenated to represent the sparse class (Equation 3), positive values correspond to the dense option, while negative values correspond to the sparse option. The complete procedure is outlined in Algorithm 1.

**Memory efficient PATCH.** To further reduce overhead, PATCH can be run in a memory-efficient manner by freezing the sparse mask parameters and optimizing only the tile-level decisions. This reduces the number of learnable parameters to $\frac{d_1 d_2}{b_1 b_2}$. While this lighter formulation limits mask-selection flexibility and can reduce performance as seen in Table 5, it makes training feasible under strict memory constraints, such as fitting an 8B model on a single 80GB GPU. We denote this version of PATCH by PATCH[Tile] and the joint optimization version of PATCH by PATCH[Joint].

## 4 EFFICIENT DEPLOYMENT OF PATCH

Executing PATCH requires handling hybrid sparse–dense tiles, a capability not supported by existing GPU libraries. Current tools either focus exclusively on dense computation (e.g., cuBLAS (NVIDIA Corporation, a), dense CUTLASS (Corporation, 2025), OpenAI Triton (Tillet et al., 2019)), or restrict support to fixed 2:4 sparsity (e.g., cuSPARSELt (NVIDIA Corporation, b), sparse CUTLASS). STOICC (Rafii et al., 2025) lifts these limitations by extending Triton with hybrid tile-level sparsity, making it a suitable backend for accelerating PATCH.

Similar to Triton, STOICC employs an inspector that benchmarks candidate kernel configurations for each sparsity ratio, identifying the most hardware-efficient tile size for the target GPU. On

---

**Algorithm 1 Joint Tile & 2:4 Mask Learning**

---

**Input:** Weight matrix $\mathbf{W}$, tile size $(b_1, b_2)$, sparsity target $\rho$, training steps $T$, loss hyperparameters $\lambda_1, \lambda_2$, temperature schedule $\{\tau_t\}_{t=1}^{T}$, scaling schedule $\{\kappa_t\}_{t=1}^{T}$.
**Output:** Learned pruning masks $\mathbf{M}^\star$, pruned weights $\widehat{\mathbf{W}}$.

1 Initialize tile logits $\mathbf{P}_{\text{tile}} \in \mathbb{R}^{\frac{d_1}{b_1} \times \frac{d_2}{b_2}}$.

2 Initialize $\mathbf{P}_{\text{tile}}$ with one-shot prior.

3 Initialize differentiable 2:4 parameters $\mathbf{P}_{2:4} \in \mathbb{R}^{6 \times \frac{d_1 d_2}{4}}$.

4 **for** $t = 1 \rightarrow T$ **do**

5      $\tilde{\mathbf{M}}_{\text{tile}} \leftarrow \text{GS}([\mathbf{P}_{\text{tile}}, 0]; \tau_t, \kappa_t)_{:,:,0} \otimes \mathbf{1}_{b_1 \times b_2}$                 ▷ Dense soft tile mask

6      $\tilde{\mathbf{M}}_{2:4} \leftarrow$ Eq. 2                                          ▷ Differentiable 2:4 mask

7      $\tilde{\mathbf{M}}_i \leftarrow \tilde{\mathbf{M}}_{\text{tile}} + (1 - \tilde{\mathbf{M}}_{\text{tile}}) \odot \tilde{\mathbf{M}}_{2:4}$               ▷ Merge masks

8      Compute loss:

$$\mathcal{L} = \mathcal{L}_{LM}(x; \tilde{\mathbf{M}} \odot \mathbf{W}) \quad + \lambda_1 \left\| \frac{\sum_i \tilde{\mathbf{M}}_i}{\sum_i \|\mathbf{W}_i\|_0} - \rho \right\|_1 \quad - \lambda_2 \frac{\sum_i \|\tilde{\mathbf{M}}_i \odot \mathbf{W}_i\|_2^2}{\sum_i \|\mathbf{W}_i\|_2^2}$$

9      Update $\mathbf{P}_{\text{tile}}, \mathbf{P}_{2:4}$ via backpropagation.

10 **end for**

11 $\mathbf{M}_{\text{tile}}^\star \leftarrow \mathbf{1}[\mathbf{P}_{\text{tile}} > 0] \otimes \mathbf{1}_{b_1 \times b_2}$                        ▷ Hard tile mask

12 $\mathbf{M}_{2:4}^\star \leftarrow$ select best 2:4 mask from $\mathbf{P}_{2:4}$.

13 $\mathbf{M}_i^\star \leftarrow \mathbf{M}_{\text{tile}}^\star + (1 - \mathbf{M}_{\text{tile}}^\star) \odot \mathbf{M}_{2:4}^\star$.

14 $\widehat{\mathbf{W}} \leftarrow \mathbf{W} \odot \mathbf{M}_i^\star$                                       ▷ Final pruned weights

**Return:** Learned mask $\mathbf{M}^\star$, pruned weights $\widehat{\mathbf{W}}$.

---

NVIDIA A100 and A6000 GPUs, our experiments show that the optimal configurations are consistently drawn from 128×128 or its subdivisions (e.g., 128×64, 64×128, 64×64). In practice, this means that regardless of the sparsity ratio or the layer shape, the chosen 128×128 granularity guarantees that STOICC's autotuned tiles can be applied consistently. Unless otherwise specified, we adopt these hardware-friendly tile sizes in all PATCH experiments. Further implementation details are provided in Appendix A.

## 5 EXPERIMENTS

**Model, dataset and evaluation.** We evaluate PATCH across diverse transformer architectures, including the Qwen-2.5 (Qwen et al., 2025), Gemma 3 (Team et al., 2025), and LLaMA-2 (Touvron et al., 2023) and 3 (Grattafiori et al., 2024) model families, spanning 500M to 8B parameters. Following the dataset size and configurations in MaskLLM (Fang et al., 2024), masks are trained for 2000 steps with a batch size of 256 on sequences with a length of 4096 tokens from the SlimPajama dataset (Soboleva et al., 2023).

Following previous LLM compression work (Mozaffari et al., 2025a; Fang et al., 2024), we evaluate the models on eight zero-shot downstream tasks: PIQA (Bisk et al., 2020), ARC-Easy and ARC-Challenge (Clark et al., 2018), Winogrande (Sakaguchi et al., 2019), OpenBookQA (Mihaylov et al., 2018), RACE (Lai et al., 2017), HellaSwag (Zellers et al., 2019), and MMLU (Hendrycks et al., 2021) using the Language Model Evaluation Harness (Gao et al., 2024) framework. Additionally, similar to previous work (Mozaffari et al., 2025a; Frantar & Alistarh, 2023; Sun et al., 2023), we evaluate the models on a language modeling task using the WikiText2 (Merity et al., 2016) dataset with a sequence length of 4096, comparing against established baselines in the following sections.

**Baselines.** To evaluate PATCH against established 2:4 sparsity pruning techniques, we compare it with the state-of-the-art learnable method MaskLLM (Fang et al., 2024), as well as one-shot methods including Wanda (Sun et al., 2023), SparseGPT (Frantar & Alistarh, 2023), Thanos (Ilin & Richtarik, 2025), ProxSparse (Liu et al., 2025) and magnitude pruning (Han et al., 2015). For one-shot pruning methods, following the default configurations in each paper, we prune the models over 128 samples from the C4 dataset.

Table 1: Model quality (average accuracy across eight zero-shot tasks and perplexity on WikiText2 dataset) for different pruning methods. By jointly optimizing the location of dense tiles and the sparsity pattern within the sparse tiles, PATCH$^{\text{Joint}}$ allows for a continuous sparsity ratio for the models, providing a flexible tradeoff between sparsity and model quality.

| Sparsity | Method | Pattern | Qwen-2.5 0.5B | | LLaMA-3.2 1B | | Gemma-3 1B | |
|---|---|---|---|---|---|---|---|---|
| | | | Acc (% ↑) | PPL (↓) | Acc (% ↑) | PPL (↓) | Acc (% ↑) | PPL (↓) |
| 0% | Dense | - | 46.00 | 12.08 | 47.70 | 9.06 | 47.01 | 11.67 |
| 50% | Magnitude | 2:4 | 30.16 | 6734.97 | 29.66 | 563.44 | 31.66 | 5005.56 |
| | Wanda | 2:4 | 32.97 | 72.48 | 31.61 | 78.18 | 34.16 | 69.41 |
| | SparseGPT | 2:4 | 34.81 | 36.59 | 35.55 | 32.73 | 35.58 | 44.59 |
| | Thanos | 2:4 | 31.31 | 37.32 | 35.71 | 33.03 | 35.09 | 62.63 |
| | ProxSparse | 2:4 | 32.05 | 111.05 | 33.55 | 49.33 | 36.63 | 90.50 |
| | MaskLLM | 2:4 | 39.33 | 15.22 | 41.04 | 12.93 | 41.84 | 12.82 |
| 45% | PATCH$^{\text{Joint}}$ | Dense/2:4 Tiles | 40.29 | 14.57 | 42.08 | 12.23 | 42.80 | 11.96 |
| 35% | PATCH$^{\text{Joint}}$ | Dense/2:4 Tiles | 41.15 | 13.84 | 42.72 | 11.67 | 43.30 | 11.48 |
| 25% | PATCH$^{\text{Joint}}$ | Dense/2:4 Tiles | 42.39 | 13.47 | 43.81 | 11.00 | 44.07 | 11.17 |

The publicly available MaskLLM pruned checkpoints are limited to LLaMA-2 7B and LLaMA-3.1 8B models. To ensure a fair comparison across all models, we implemented MaskLLM in PyTorch and replicated its results for additional architectures presented in this study.

We faced a similar challenge with ProxSparse as well, where only the LLaMA-2-7B and LLaMA-3.1-8B checkpoints are publicly available. We have pruned other models with their official code base using their default hyperparameters for comparison.

Additional implementation details and hyperparameters used in our experiments are provided in Appendix D.

## 5.1 MODEL QUALITY RESULTS

**Joint sparse and dense tile optimization.** For smaller models like Qwen-2.5 0.5B, LLaMA-3.2 1B, and Gemma-3 1B, we apply the joint variant PATCH$^{\text{Joint}}$, which simultaneously optimizes dense tile locations and sparsity patterns within sparse tiles. This approach enables effective performance.

The average accuracy of the models across eight zero-shot downstream tasks and their perplexity on the WikiText2 dataset is reported in Table 1. The results demonstrate that PATCH$^{\text{Joint}}$ provides a flexible tradeoff between sparsity ratio and model quality, narrowing the performance gap to dense models while ensuring hardware-friendly inference. A similar pattern holds for larger models using a memory-efficient variant, as explored next.

**Memory-efficient tile selection.** For larger models such as LLaMA-2 7B and LLaMA-3.1 8B, we employ the memory-efficient variant PATCH$^{\text{Tile}}$, which freezes the fine-grained sparse weight structure while optimizing dense tile selections.

Table 2 summarizes the average accuracy of the models across eight downstream tasks in addition to their perplexity on the WikiText2 dataset for different sparsity ratios, illustrating that PATCH$^{\text{Tile}}$ delivers a comparable flexible sparsity-quality tradeoff when using a high-quality frozen 2:4 mask.

Overall, across Tables 1 and 2, PATCH consistently surpasses one-shot methods like Wanda, SparseGPT, and magnitude pruning due to its end-to-end training on large corpora. While MaskLLM also trains end-to-end on a large dataset, its fixed 2:4 sparsity ratio limits achievable accuracy and perplexity. In contrast, PATCH overcomes this limitation with flexible dense tile allocation, achieving accuracy gains and perplexity reductions from 45% to 25% sparsity that progressively align with dense model performance. The full per-task accuracy results are provided in Appendix B.

Table 2: Model quality (average accuracy across eight zero-shot tasks and perplexity on WikiText2 dataset) for different pruning methods. By only optimizing the location of dense tiles while keeping sparsity pattern within the sparse tiles frozen, PATCH$^{\text{Tile}}$ provides a memory efficient variant for PATCH$^{\text{Joint}}$, allowing for a continuous sparsity ratio for the models and providing a flexible tradeoff between sparsity and model quality.

| Sparsity | Method | Pattern | LLaMA-2 7B | | LLaMA-3.1 8B | |
|---|---|---|---|---|---|---|
| | | | Acc (% ↑) | PPL (↓) | Acc (% ↑) | PPL (↓) |
| 0% | Dense | - | 54.61 | 5.12 | 60.31 | 5.84 |
| 50% | Magnitude | 2:4 | 43.44 | 54.39 | 35.93 | 765.92 |
| | Wanda | 2:4 | 44.30 | 11.15 | 41.77 | 21.29 |
| | SparseGPT | 2:4 | 45.09 | 10.12 | 45.53 | 15.11 |
| | Thanos | 2:4 | 44.80 | 11.19 | 45.72 | 16.09 |
| | ProxSparse | 2:4 | 45.92 | 9.18 | 45.14 | 15.17 |
| | MaskLLM | 2:4 | 48.62 | 6.78 | 52.80 | 8.58 |
| 45% | PATCH$^{\text{Tile}}$ | Dense/2:4 Tiles | 48.99 | 6.55 | 53.60 | 8.20 |
| 35% | PATCH$^{\text{Tile}}$ | Dense/2:4 Tiles | 50.08 | 6.18 | 55.28 | 7.89 |
| 25% | PATCH$^{\text{Tile}}$ | Dense/2:4 Tiles | 51.58 | 5.86 | 56.48 | 7.34 |

Table 3: Impact of PATCH's tile size across sparsity levels (↓ is better). The effect of tile size on model quality is not significant, showing PATCH's robustness against tile size.

Table 4: Global sparsity yields better quality by concentrating pruning in less important blocks and preserving density elsewhere (↓ is better).

| Sparsity (0.5B) | 128 | 64 | 32 | 16 | 8 | 4 |
|---|---|---|---|---|---|---|
| 45% | 14.57 | 14.66 | 14.70 | 14.67 | 14.70 | **14.55** |
| 35% | 13.84 | 14.08 | 14.15 | 14.03 | 14.01 | **13.72** |
| 25% | 13.47 | 13.54 | 13.52 | 13.53 | 13.40 | **13.11** |

| Sparsity (0.5B) | Global | Layer-wise |
|---|---|---|
| 45% | **14.57** | 15.17 |
| 35% | **13.84** | 14.48 |
| 25% | **13.47** | 13.95 |

## 5.2 Understanding the components of PATCH

This subsection examines the design choices driving PATCH's performance by analyzing its behavior across various configurations on the Qwen-2.5 0.5B model.

**Tile size.** We initially assess the impact of tile size on PATCH's performance, fixing hyperparameters to those optimized for 128×128 tiles. Table 3 reveals that $4 \times 4$ tiles maximize model quality through finer sparse-dense control, though larger tile sizes show minimal variation, suggesting robustness. However, smaller tiles may hinder hardware efficiency, requiring a balance with hardware specifications.

**Joint vs. tile-only mask search.** We then analyze the impact of fixing the 2:4 masks and optimizing only tile masks. Table 5 shows that among frozen 2:4 masks, MaskLLM provides the strongest results. On the other hand, one-shot pruning methods perform comparably at higher sparsity levels but diverge at lower sparsity, with SparseGPT emerging as the best overall. When comparing against our full approach, joint optimization of both tile and 2:4 masks consistently outperforms tile-only training across sparsity ratios. Nevertheless, tile-only training remains a practical alternative for larger models in resource-constrained settings, as also reflected in Table 2.

**Sparsity allocation.** We analyze how sparsity is allocated across transformer blocks under a global target. Across models, deeper transformer blocks are pruned far less, while the initial blocks also tend to receive lighter pruning depending on the architecture. By contrast, the middle blocks consistently absorb most of the sparsity, suggesting that they contain more redundancy (Figure 2). We compare this flexible allocation to enforcing sparsity uniformly at the layer level. As shown in Table 4, global targets deliver better results by pruning more aggressively in redundant layers while

Table 5: Impact of fixed 2:4 mask selection for PATCH$^{Tile}$, compared with joint optimization ($\downarrow$ is better). PATCH$^{Joint}$ achieves the lowest perplexity overall, while for PATCH$^{Tile}$, MaskLLM provides the best frozen mask.

| Sparsity (0.5B) | MaskLLM | SparseGPT (w/o weight update) | Wanda | Magnitude | PATCH$^{Joint}$ |
|---|---|---|---|---|---|
| 45% | **15.06** | 21.84 | 21.83 | 21.33 | 14.57 |
| 35% | **14.55** | 17.29 | 17.96 | 19.90 | 13.84 |
| 25% | **14.17** | 14.89 | 15.09 | 16.05 | 13.47 |

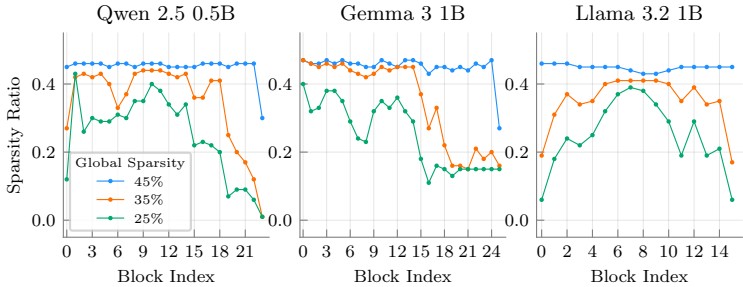

Figure 2: Layer-wise sparsity allocation under different global sparsity budgets for various models. PATCH achieves the target global sparsity while flexibly distributing pruning across transformer layers.

preserving capacity in sensitive ones. In contrast, layer-wise targets impose uniform sparsity that can over-prune critical components (Li et al., 2024b; Xu et al., 2024; Li et al., 2024a; Yin et al., 2025).

On top of variation across depth, sparsity is also distributed unevenly across the individual linear layers within each transformer block. Figure 3 breaks down the allocation into the query, key, value, and output matrices of the attention module, as well as the up, gate, and down matrices of the MLP for the Qwen 2.5 0.5B model. The up, gate, and down layers absorb most of the sparsity and largely explain the overall allocation pattern seen in Figure 2. In contrast, the attention module is treated as more critical. The key and value matrices are never pruned, while the output matrix shows moderate pruning at higher global sparsity targets. The query matrix is pruned the most, suggesting it is the least important within the attention submodule. The distributions for the Gemma-3-1B and LLaMA-3.2-1B models are provided in Appendix E, where the same pattern is observed.

## 5.3 COMBINATION WITH OTHER COMPRESSION METHODS

LLM compression relies on three orthogonal methods—sparsity, quantization, and low-rank approximation—which can be combined. While this work focuses on sparsity, this section demonstrates how PATCH integrates with these other techniques.

**Quantization.** Quantization reduces memory and accelerates computation by lowering numerical precision on hardware optimized for low bitwidths.

**Low-rank approximation.** Low-rank methods complement sparsity and quantization by reintroducing a small number of parameters to recover accuracy, with SLiM (Mozaffari et al., 2025a) as a leading one-shot technique.

Table 6 reports results on LLaMA-2 7B and LLaMA-3.1 8B, comparing PATCH and MaskLLM under 4-bit weight-only quantization, as well as an additional setting that combines our method with a low-rank adapter (of 10% of the weight's rank). These results show that sparsity, quantization, and low-rank approximation can be composed to achieve controllable tradeoffs between compression and model quality, and that our approach integrates seamlessly with both techniques within broader compression pipelines.

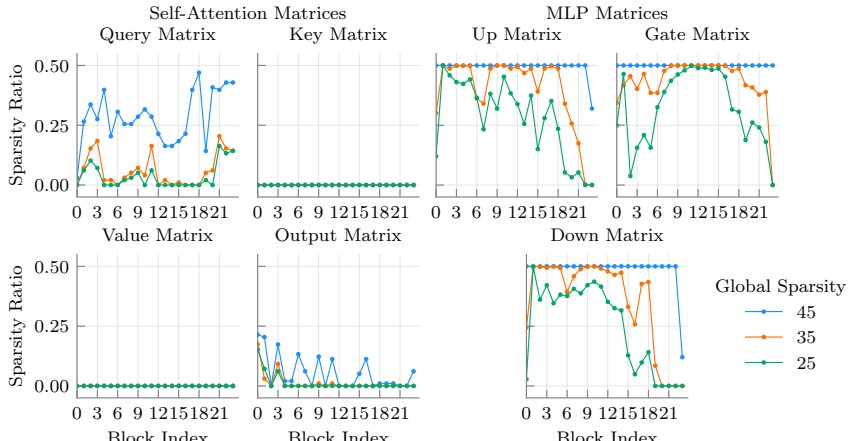

Figure 3: Sparsity distribution across Attention and MLP layers under varying global sparsity budgets in Qwen-2.5 0.5B.

Table 6: Average accuracy (↑ indicates better) across eight zero-shot downstream tasks and WikiText2 perplexity (↓ indicates better) of compressed models with **4-bit weight-only quantization**. Please note that using LoRA adds additional parameters to the model. *Comp. Ratio* refers to the theoretical weight memory compression factor relative to the dense model.

| Sparsity | Method | Bit | LoRA | LLaMA-2-7B | | LLaMA-3.1-8B | | Comp. |
|---|---|---|---|---|---|---|---|---|
| | | | | Acc (% ↑) | PPL (↓) | Acc (% ↑) | PPL (↓) | Ratio |
| 0% | Dense | - | - | 54.61 | 5.12 | 60.31 | 5.84 | 1x |
| 50% | MaskLLM | 4 | - | 47.98 | 7.64 | 51.12 | 9.92 | 5.33x |
| 45% | PATCH$^{\text{Tile}}$ | 4 | - | 48.19 | 7.34 | 52.47 | 9.68 | 5.16x |
| 45% | PATCH$^{\text{Tile}}$ | 4 | SLIM-LoRA | 50.71 | 6.83 | 54.04 | 9.12 | 4.10x |
| 35% | PATCH$^{\text{Tile}}$ | 4 | - | 49.38 | 6.92 | 53.81 | 9.26 | 4.85x |
| 35% | PATCH$^{\text{Tile}}$ | 4 | SLIM-LoRA | 51.91 | 6.42 | 55.70 | 8.37 | 3.90x |
| 25% | PATCH$^{\text{Tile}}$ | 4 | - | 50.45 | 6.57 | 55.45 | 8.69 | 4.57x |
| 25% | PATCH$^{\text{Tile}}$ | 4 | SLIM-LoRA | 52.62 | 6.11 | 56.99 | 7.77 | 3.72x |

## 5.4 SPEEDUP AND MEMORY SAVINGS

We evaluate the inference efficiency of the LLaMA-2 7B model pruned with PATCH using the STOICC (Rafii et al., 2025) compiler. With a batch size of 16 on an A6000 GPU, we observe end-to-end throughput improvements of $1.18\times$, $1.27\times$, and $1.38\times$ at sparsity levels of 25%, 35%, and 45%, respectively, compared to the dense baseline. At the same sparsity levels, the model's GPU memory footprint during inference is also reduced, dropping to $0.76\times$, $0.68\times$, and $0.59\times$ of the fully dense model, respectively. These results underscore the trade-off between accuracy retention and the computational savings enabled by sparsity.

## 6 CONCLUSION

We introduced PATCH, a hybrid sparsity framework that bridges the gap between unstructured and 2:4 sparsity for large language models. By partitioning weight matrices into tiles designated as either dense or 2:4 sparse, PATCH enables adaptive sparsity ratios between 0% and 50%, balancing accuracy and acceleration.

Experiments across models up to 8B parameters show that PATCH consistently improves accuracy over state-of-the-art 2:4 pruning methods while achieving up to $1.38\times$ end-to-end speedup on consumer grade GPUs. These results demonstrate the promise of hybrid sparsity as a practical approach

to efficient LLM inference and motivate future work on broader sparsity formats, integration with quantization, and co-design with hardware kernels.

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

# A STOICC INTEGRATION

Triton (Tillet et al., 2019) enables developers to write efficient GPU kernels with a Python-like syntax, but it natively supports only dense matrix operations and cannot handle sparsity. To accelerate the mixed-tile format produced by PATCH, we employ the STOICC compiler (Rafii et al., 2025). STOICC extends Triton with a sparse code-generation backend that allows tiles within a matrix to be either dense or sparse, enabling mixed execution within a single matrix multiplication.

We rely on STOICC's inspector to autotune both tile sizes and execution schedules (i.e., alternative kernel execution schemes such as split-$K$ parallelism) for the prefill and decoding stages of LLM inference. Matrix compression and metadata generation are determined by the chosen tile size, which must remain consistent across both stages. To address this, we first autotune the decoding stage, which is the primary bottleneck of autoregressive generation, since it is executed once per generated token (e.g., 128 times for 128 new tokens), unlike the single pass of prefill. The optimal tile size identified for decoding are then fixed and reused for prefill, where we perform a second round of autotuning over the remaining independent parameters.

In contrast, for fully 2:4 sparse matrices, compression is independent of the block size, so they can be autotuned in the same way as dense kernels in Triton without this coupling constraint.

The pseudocode outlining this process, including the handling of dense, fully 2:4 sparse, and mixed-sparsity modules, is provided in PseudoCode 1.

```
1  def tune_and_convert_model(M, backend_name):
2      // backend_name  ∈  {"STOICC", "cuSPARSELt"}
3      2_4_backend = select_2_4_backend(backend_name)
4
5      // create all configs & schedules to tune over
6      base_configs = STOICC.create_configs()
7      inspector    = Inspector()
8
9      for each module in M:
10         s = get_sparsity_ratio(module.weight)
11
12         // Keep dense Torch (cuBLAS) module
13         if s == 0:
14             continue
15
16         // Use STOICC or cuSPARSELt for fully 2:4
17         elif s == 0.5:
18             c          = 2_4_backend.compress(module.weight)
19             new_module = 2_4_backend.create_module(c)
20             replace(module, new_module)
21             continue
22
23         else:
24             decoding_input = Tensor(BS, module.weight.shape[1])
25             prefill_input  = Tensor(BS * SL, module.weight.shape[1])
26
27             // Tune on decoding input first
28             inspector.set_configs(base_configs)
29             best_cfg_dec = inspector.inspect(
30                 decoding_input,
31                 module.weight,
32                 isASparse=False)
33             BN = best_cfg_dec["BLOCK_N"]
34             BK = best_cfg_dec["BLOCK_K"]
35
36             // Tune on prefill using decoding tile sizes
37             prefill_cfg = STOICC.create_configs(BLOCK_N=BN, BLOCK_K=BK)
38             inspector.set_configs(prefill_cfg)
39             best_cfg_pre = inspector.inspect(
40                 prefill_input,
41                 module.weight,
```

```
42              isASparse=False)
43
44          c = inspector.compress(module.weight, BN, BK)
45          mixed_module = MixedModule(c, best_cfg_dec, best_cfg_pre)
46          replace(module, mixed_module)
47
48      return M
```

PseudoCode 1: Tuning and Converting Model Weights to Mixed Format.

Table 7 reports the measured throughput (tokens processed per second) of LLaMA-2 7B at sparsity levels of 45%, 35%, and 25% with a batch size of 16 on an A6000 GPU. To reduce CPU overhead from launching Triton kernels in PyTorch, we executed generation through CUDA graphs, capturing both the prefill and decoding stages. With sparsity ratios between 25% and 45%, our heterogeneous approach achieves $1.18\times$–$1.38\times$ end-to-end acceleration over the dense baseline. We also report timings on A100 in Table 8.

Table 7: Throughput of LLaMA-2 7B with mixed sparsity compared to the dense model. Measurements taken on an A6000 GPU with batch size 16. Throughput is reported in tokens processed/sec.

| Sparsity | Prefill length | Tokens generated | Throughput (tok/s) | Speedup vs. dense |
|---|---|---|---|---|
| 0% | 128 | 128 | 1023.80 | $1.00\times$ |
| 25% | 128 | 128 | 1212.79 | $1.18\times$ |
| 35% | 128 | 128 | 1304.46 | $1.27\times$ |
| 45% | 128 | 128 | 1410.20 | $1.38\times$ |
| 0% | 128 | 1024 | 435.42 | $1.00\times$ |
| 25% | 128 | 1024 | 493.33 | $1.13\times$ |
| 35% | 128 | 1024 | 515.39 | $1.18\times$ |
| 45% | 128 | 1024 | 542.87 | $1.25\times$ |

## B    PER TASK RESULTS

This appendix provides detailed per-task accuracy results for the models evaluated in Section 5, covering eight zero-shot downstream tasks: MMLU, PIQA, ARC-Easy, ARC-Challenge, Winogrande, OpenbookQA, RACE, and Hellaswag. The results are presented for each model at various sparsity levels and pruning methods, including our proposed PATCH[Joint] and PATCH[Tile] variants, alongside baseline methods such as Magnitude, Wanda, SparseGPT, Thanos, ProxSparse, and MaskLLM. These tables complement the average accuracy and perplexity results reported in Tables 1 and 2 of the main paper, offering a granular view of model performance across individual tasks.

For smaller models (Qwen-2.5 0.5B, LLaMA-3.2 1B, and Gemma-3 1B), we report results using the PATCH[Joint] variant, which jointly optimizes dense tile locations and sparsity patterns within

Table 8: Throughput of LLaMA-2 7B with mixed sparsity compared to the dense model. Measurements taken on an A100 GPU with batch size 16. Throughput is reported in tokens processed/sec.

| Sparsity | Prefill length | Tokens generated | Throughput (tok/s) | Speedup vs. dense |
|---|---|---|---|---|
| 0% | 128 | 128 | 1876.24 | $1.00\times$ |
| 25% | 128 | 128 | 2002.02 | $1.07\times$ |
| 35% | 128 | 128 | 2088.98 | $1.11\times$ |
| 45% | 128 | 128 | 2180.88 | $1.16\times$ |
| 0% | 128 | 1024 | 812.55 | $1.00\times$ |
| 25% | 128 | 1024 | 864.66 | $1.06\times$ |
| 35% | 128 | 1024 | 885.90 | $1.09\times$ |
| 45% | 128 | 1024 | 907.12 | $1.12\times$ |

sparse tiles. For larger models (LLaMA-2 7B and LLaMA-3.1 8B), we report results using the memory-efficient PATCH$^{\text{Tile}}$ variant, which optimizes dense tile selections with a fixed 2:4 sparsity mask. The per-task accuracies highlight the effectiveness of our approaches in maintaining robust performance across diverse tasks, even at high sparsity levels, compared to baseline methods.

The following tables detail the per-task accuracies for each model:

- **Qwen-2.5 0.5B**: Table 9 presents the per-task accuracies for the PATCH$^{\text{Joint}}$ variant and baselines at 0% and 50% sparsity, with PATCH$^{\text{Joint}}$ evaluated at 25%, 35%, and 45% sparsity.

- **LLaMA-2 7B**: Table 10 shows the per-task accuracies for the PATCH$^{\text{Tile}}$ variant and baselines, with PATCH$^{\text{Tile}}$ evaluated at 25%, 35%, and 45% sparsity.

- **LLaMA-3.1 8B**: Table 11 provides the per-task accuracies for the PATCH$^{\text{Tile}}$ variant and baselines, with PATCH$^{\text{Tile}}$ at 25%, 35%, and 45% sparsity.

- **LLaMA-3.2 1B**: Table 12 reports the per-task accuracies for the PATCH$^{\text{Joint}}$ variant and baselines, with PATCH$^{\text{Joint}}$ at 25%, 35%, and 45% sparsity.

- **Gemma-3 1B**: Table 13 details the per-task accuracies for the PATCH$^{\text{Joint}}$ variant and baselines, with PATCH$^{\text{Joint}}$ at 25%, 35%, and 45% sparsity.

These results enable a deeper analysis of the task-specific performance trends, demonstrating the flexibility and robustness of PATCH$^{\text{Joint}}$ and PATCH$^{\text{Tile}}$ in achieving high accuracy across diverse tasks while maintaining hardware-friendly sparsity patterns.

Table 9: Model quality (task accuracy across eight zero-shot tasks, reported in %) for Qwen-2.5 0.5B with different pruning methods. PATCH$^{\text{Joint}}$ optimizes dense tile locations and sparsity patterns, enabling a flexible sparsity-quality tradeoff.

| Sparsity | Method | Pattern | MMLU | PIQA | ARC-E | ARC-C | WinoG. | OBQA | RACE | HellaS. | Avg |
|---|---|---|---|---|---|---|---|---|---|---|---|
| 0% | Dense | - | 47.71 | 70.24 | 64.48 | 29.52 | 56.20 | 24.20 | 35.02 | 40.63 | 46.00 |
| 50% | Magnitude | 2:4 | 23.00 | 54.24 | 31.23 | 19.20 | 49.96 | 13.60 | 23.44 | 26.59 | 30.16 |
| | Wanda | 2:4 | 24.43 | 58.71 | 43.18 | 17.75 | 51.62 | 12.20 | 26.32 | 29.58 | 32.97 |
| | SparseGPT | 2:4 | 22.93 | 60.77 | 46.60 | 20.82 | 52.88 | 14.00 | 29.57 | 30.93 | 34.81 |
| | Thanos | 2:4 | 22.97 | 60.17 | 45.37 | 19.20 | 53.59 | 15.20 | 31.00 | 31.31 | 34.85 |
| | ProxSparse | 2:4 | 23.00 | 57.34 | 40.53 | 18.26 | 48.62 | 14.00 | 25.65 | 29.02 | 32.05 |
| | MaskLLM | 2:4 | 25.11 | 67.03 | 56.57 | 23.98 | 52.57 | 20.20 | 33.30 | 35.90 | 39.33 |
| 45% | PATCH$^{\text{Joint}}$ | Dense/2:4 Tiles | 27.39 | 68.44 | 59.13 | 25.77 | 53.67 | 19.80 | 32.15 | 35.99 | 40.29 |
| 35% | PATCH$^{\text{Joint}}$ | Dense/2:4 Tiles | 29.04 | 68.88 | 60.40 | 26.37 | 55.09 | 20.40 | 32.44 | 36.58 | 41.15 |
| 25% | PATCH$^{\text{Joint}}$ | Dense/2:4 Tiles | 30.89 | 69.15 | 62.79 | 29.10 | 55.33 | 20.00 | 34.16 | 37.71 | 42.39 |

Table 10: Model quality (task accuracy across eight zero-shot tasks, reported in %) for LLaMA-2 7B with different pruning methods. PATCH$^{\text{Tile}}$ optimizes tile-based sparsity, enabling a flexible sparsity-quality tradeoff.

| Sparsity | Method | Pattern | MMLU | PIQA | ARC-E | ARC-C | WinoG. | OBQA | RACE | HellaS. | Avg |
|---|---|---|---|---|---|---|---|---|---|---|---|
| 0% | Dense | - | 41.82 | 78.07 | 76.35 | 43.52 | 69.06 | 31.40 | 39.52 | 57.13 | 54.61 |
| 50% | Magnitude | 2:4 | 25.82 | 70.02 | 61.78 | 30.12 | 61.01 | 21.80 | 31.48 | 45.45 | 43.44 |
| | Wanda | 2:4 | 25.80 | 71.00 | 63.80 | 30.29 | 61.09 | 25.20 | 35.50 | 41.75 | 44.30 |
| | SparseGPT | 2:4 | 26.17 | 70.73 | 63.80 | 30.63 | 65.04 | 24.00 | 37.13 | 43.18 | 45.09 |
| | Thanos | 2:4 | 25.27 | 70.78 | 63.43 | 30.97 | 64.56 | 23.80 | 36.46 | 43.11 | 44.80 |
| | ProxSparse | 2:4 | 26.77 | 71.60 | 65.70 | 33.02 | 62.90 | 24.20 | 35.31 | 47.84 | 45.92 |
| | MaskLLM | 2:4 | 27.65 | 74.76 | 69.44 | 35.58 | 65.04 | 26.80 | 38.56 | 51.15 | 48.62 |
| 45% | PATCH$^{\text{Tile}}$ | Dense/2:4 Tiles | 27.28 | 75.41 | 70.16 | 35.84 | 65.27 | 27.60 | 38.76 | 51.61 | 48.99 |
| 35% | PATCH$^{\text{Tile}}$ | Dense/2:4 Tiles | 29.93 | 76.71 | 70.88 | 36.95 | 65.67 | 28.20 | 39.33 | 52.96 | 50.08 |
| 25% | PATCH$^{\text{Tile}}$ | Dense/2:4 Tiles | 32.33 | 76.99 | 72.81 | 38.57 | 68.27 | 29.80 | 39.52 | 54.34 | 51.58 |

Table 11: Model quality (task accuracy across eight zero-shot tasks, reported in %) for LLaMA-3.1 8B with different pruning methods. PATCH$^{\text{Tile}}$ optimizes tile-based sparsity, enabling a flexible sparsity-quality tradeoff.

| Sparsity | Method | Pattern | MMLU | PIQA | ARC-E | ARC-C | WinoG. | OBQA | RACE | HellaS. | Avg |
|---|---|---|---|---|---|---|---|---|---|---|---|
| 0% | Dense | - | 63.57 | 80.09 | 81.44 | 51.37 | 73.48 | 33.40 | 39.14 | 60.02 | 60.31 |
| 50% | Magnitude | 2:4 | 23.06 | 63.82 | 45.33 | 25.94 | 53.91 | 15.20 | 26.70 | 33.49 | 35.93 |
| | Wanda | 2:4 | 27.85 | 68.88 | 58.33 | 26.71 | 60.93 | 19.00 | 33.78 | 38.70 | 41.77 |
| | SparseGPT | 2:4 | 31.82 | 70.46 | 63.85 | 31.74 | 64.56 | 21.60 | 37.22 | 42.99 | 45.53 |
| | Thanos | 2:4 | 34.23 | 70.40 | 63.13 | 31.40 | 63.61 | 23.20 | 37.03 | 42.75 | 45.72 |
| | ProxSparse | 2:4 | 29.89 | 71.71 | 62.63 | 33.28 | 58.56 | 23.80 | 35.22 | 46.03 | 45.14 |
| | MaskLLM | 2:4 | 42.47 | 77.04 | 73.15 | 40.19 | 68.43 | 28.80 | 38.28 | 54.04 | 52.80 |
| 45% | PATCH$^{\text{Tile}}$ | Dense/2:4 Tiles | 47.32 | 77.96 | 73.61 | 41.89 | 68.03 | 29.00 | 36.56 | 54.44 | 53.60 |
| 35% | PATCH$^{\text{Tile}}$ | Dense/2:4 Tiles | 51.15 | 77.97 | 76.14 | 42.41 | 69.46 | 31.40 | 38.18 | 55.54 | 55.28 |
| 25% | PATCH$^{\text{Tile}}$ | Dense/2:4 Tiles | 52.95 | 77.75 | 77.57 | 44.62 | 70.56 | 31.80 | 39.90 | 56.69 | 56.48 |

Table 12: Model quality (task accuracy across eight zero-shot tasks, reported in %) for LLaMA-3.2 1B with different pruning methods. PATCH$^{\text{Joint}}$ optimizes dense tile locations and sparsity patterns, enabling a flexible sparsity-quality tradeoff.

| Sparsity | Method | Pattern | MMLU | PIQA | ARC-E | ARC-C | WinoG. | OBQA | RACE | HellaS. | Avg |
|---|---|---|---|---|---|---|---|---|---|---|---|
| 0% | Dense | - | 37.57 | 74.54 | 65.53 | 31.32 | 60.62 | 26.40 | 37.89 | 47.76 | 47.70 |
| 50% | Magnitude | 2:4 | 23.31 | 53.81 | 27.74 | 18.94 | 51.38 | 11.80 | 24.02 | 26.26 | 29.66 |
| | Wanda | 2:4 | 22.90 | 58.11 | 37.08 | 19.20 | 49.09 | 13.20 | 25.17 | 28.11 | 31.61 |
| | SparseGPT | 2:4 | 22.93 | 61.43 | 45.03 | 22.35 | 54.93 | 15.80 | 29.86 | 32.08 | 35.55 |
| | Thanos | 2:4 | 23.12 | 62.40 | 44.91 | 21.76 | 54.30 | 16.00 | 31.10 | 32.09 | 35.71 |
| | ProxSparse | 2:4 | 22.96 | 60.83 | 39.44 | 20.31 | 51.54 | 16.80 | 25.17 | 31.37 | 33.55 |
| | MaskLLM | 2:4 | 26.28 | 69.10 | 57.41 | 25.85 | 55.48 | 21.40 | 32.82 | 39.94 | 41.04 |
| 45% | PATCH$^{\text{Joint}}$ | Dense/2:4 Tiles | 23.81 | 70.89 | 60.77 | 27.22 | 56.27 | 22.80 | 34.07 | 40.78 | 42.08 |
| 35% | PATCH$^{\text{Joint}}$ | Dense/2:4 Tiles | 25.13 | 71.32 | 60.27 | 29.18 | 57.06 | 22.00 | 34.64 | 42.17 | 42.72 |
| 25% | PATCH$^{\text{Joint}}$ | Dense/2:4 Tiles | 28.59 | 71.44 | 61.57 | 28.67 | 58.25 | 23.20 | 35.22 | 43.52 | 43.81 |

Table 13: Model quality (accuracy across eight zero-shot tasks) for Gemma-3 1B with different pruning methods. PATCH$^{\text{Joint}}$ optimizes dense tile locations and sparsity patterns, enabling a flexible sparsity-quality tradeoff.

| Sparsity | Method | Pattern | MMLU | PIQA | ARC-E | ARC-C | WinoG. | OBQA | RACE | HellaS. | Avg |
|---|---|---|---|---|---|---|---|---|---|---|---|
| 0% | Dense | - | 24.95 | 75.03 | 71.84 | 34.90 | 58.64 | 28.60 | 34.83 | 47.26 | 47.01 |
| 50% | Magnitude | 2:4 | 23.08 | 59.79 | 37.29 | 17.66 | 50.59 | 14.00 | 22.87 | 27.97 | 31.66 |
| | Wanda | 2:4 | 23.96 | 59.52 | 48.02 | 18.34 | 51.22 | 14.20 | 27.85 | 30.18 | 34.16 |
| | SparseGPT | 2:4 | 23.62 | 62.79 | 49.83 | 19.03 | 51.54 | 15.20 | 30.62 | 31.99 | 35.58 |
| | Thanos | 2:4 | 23.44 | 62.24 | 48.86 | 18.34 | 50.12 | 15.60 | 30.81 | 31.28 | 35.09 |
| | ProxSparse | 2:4 | 23.10 | 64.25 | 50.72 | 21.59 | 53.43 | 18.00 | 29.09 | 32.86 | 36.63 |
| | MaskLLM | 2:4 | 25.03 | 69.91 | 60.27 | 27.65 | 56.27 | 21.20 | 34.55 | 39.84 | 41.84 |
| 45% | PATCH$^{\text{Joint}}$ | Dense/2:4 Tiles | 23.54 | 71.65 | 63.97 | 27.47 | 57.30 | 23.60 | 33.49 | 41.39 | 42.80 |
| 35% | PATCH$^{\text{Joint}}$ | Dense/2:4 Tiles | 25.38 | 72.31 | 63.80 | 27.39 | 56.67 | 24.00 | 34.74 | 42.07 | 43.30 |
| 25% | PATCH$^{\text{Joint}}$ | Dense/2:4 Tiles | 25.45 | 71.87 | 66.16 | 30.55 | 57.85 | 22.80 | 34.55 | 43.33 | 44.07 |

## C    TILE TRANSFER LEARNING

We also test whether initializing tile logits with priors from one-shot pruning methods improves performance, as done in MaskLLM (Fang et al., 2024). In our case, the initialization is derived from one-shot pruning with unstructured sparsity. We initialize tiles that retain more nonzeros after unstructured pruning with positive logits (favoring dense assignment), while the remaining tiles receive negative logits, controlled by a strength parameter. The number of tiles initialized as dense is selected such that the overall layer-wise sparsity target is satisfied. As shown in Table 14, the

choice of prior has little impact on final performance: all priors yield nearly identical perplexity, with random initialization often performing best. This is likely because the global sparsity target enables dynamic reallocation of sparsity across layers during training, overriding the effect of any fixed initialization. For consistency with prior work, we adopt SparseGPT initialization in all experiments.

Table 14: Perplexity ($\downarrow$) under different tile prior initializations. All priors yield nearly identical performance, suggesting that the global sparsity target allows dynamic reallocation of sparsity during training, overriding the influence of fixed initialization.

| Sparsity (0.5B) | Nothing | SparseGPT | Wanda | Magnitude | Random |
|---|---|---|---|---|---|
| 45% | 14.80 | 14.57 | 14.50 | **14.48** | 14.51 |
| 35% | 13.97 | 13.84 | 13.87 | 13.85 | **13.79** |
| 25% | 13.47 | 13.47 | 13.37 | 13.44 | **13.33** |

## D  IMPLEMENTATION DETAILS AND HYPERPARAMETERS

We train all masks using the HuggingFace Trainer API (Wolf et al., 2020) for 2000 steps with a global batch size of 256 and a sequence length of 4096, resulting in 2B tokens processed from the SlimPajama corpus (Soboleva et al., 2023).

Training is accelerated via data parallelism across a single node with 4 H100 GPUs. In this setup, PATCH$^{\text{Joint}}$ requires 18 and 24 GPU hours on the 0.5B and 1B models, respectively, while PATCH$^{\text{Tile}}$ requires 84 and 96 GPU hours on the 7B and 8B models.

The hyperparameters for PATCH$^{\text{Joint}}$ and PATCH$^{\text{Tile}}$ are summarized in Table 15, tuned on Qwen-2.5-0.5B. For the 2:4 mask parameters, we follow the configuration from MaskLLM (Fang et al., 2024).

Table 15: Hyper-parameters used for PATCH$^{\text{Joint}}$ and PATCH$^{\text{Tile}}$ across sparsity ratios. All hyper parameters were tuned on Qwen-2.5-0.5B.

| Sparsity | Method | Optimizer | Logits Init | Gumbel Scaling | Gumbel | Prior(Strength) | Sparse Reg. | Weight Reg. |
|---|---|---|---|---|---|---|---|---|
| 25% | PATCH$^{\text{Joint}}$ | Adam(0.001) | $\mathcal{N}(0, 0.014)$ | $25 \to 350$ | $2 \to 0.05$ | SparseGPT(3) | 7 | 10 |
| 35% | PATCH$^{\text{Joint}}$ | Adam(0.001) | $\mathcal{N}(0, 0.014)$ | $25 \to 350$ | $2 \to 0.05$ | SparseGPT(3) | 7 | 10 |
| 45% | PATCH$^{\text{Joint}}$ | Adam(0.001) | $\mathcal{N}(0, 0.014)$ | $25 \to 350$ | $4 \to 0.05$ | SparseGPT(3) | 7 | 10 |
| 25% | PATCH$^{\text{Tile}}$ | Adam(0.0001) | $\mathcal{N}(0, 0.014)$ | $100 \to 500$ | $2 \to 0.05$ | SparseGPT(3) | 3 | 0.1 |
| 35% | PATCH$^{\text{Tile}}$ | Adam(0.0001) | $\mathcal{N}(0, 0.014)$ | $100 \to 500$ | $2 \to 0.05$ | SparseGPT(3) | 3 | 0.1 |
| 45% | PATCH$^{\text{Tile}}$ | Adam(0.0001) | $\mathcal{N}(0, 0.014)$ | $100 \to 500$ | $2 \to 0.05$ | SparseGPT(3) | 3 | 0.1 |

## E  ADDITIONAL LAYER-WISE SPARSITY DISTRIBUTIONS

In this appendix, we provide the sparsity distributions for the Gemma-3-1B (Figure 4) and Llama-3.2-1B (Figure 5) models, as referenced in the main text. Similar to the Qwen-2.5 0.5B model, the patterns observed here indicate that MLP layers (up, gate, and down matrices) are pruned more aggressively, absorbing the majority of sparsity. In contrast, the self-attention layers are treated as more critical, with key and value matrices remaining largely dense or unpruned, while the query matrix experiences the highest pruning within the attention submodule, and the output matrix shows moderate pruning under higher global sparsity targets. This consistent behavior across models underscores the redundancy in MLP components and the sensitivity of attention mechanisms.

## F  RELATED WORK

### F.1  PRUNING METHODS

Pruning is one of the most widely studied approaches for compressing deep neural networks, with the goal of removing redundant parameters while preserving accuracy. Classical pruning methods can be broadly categorized into *local* (layer-wise) and *global* (end-to-end) strategies.

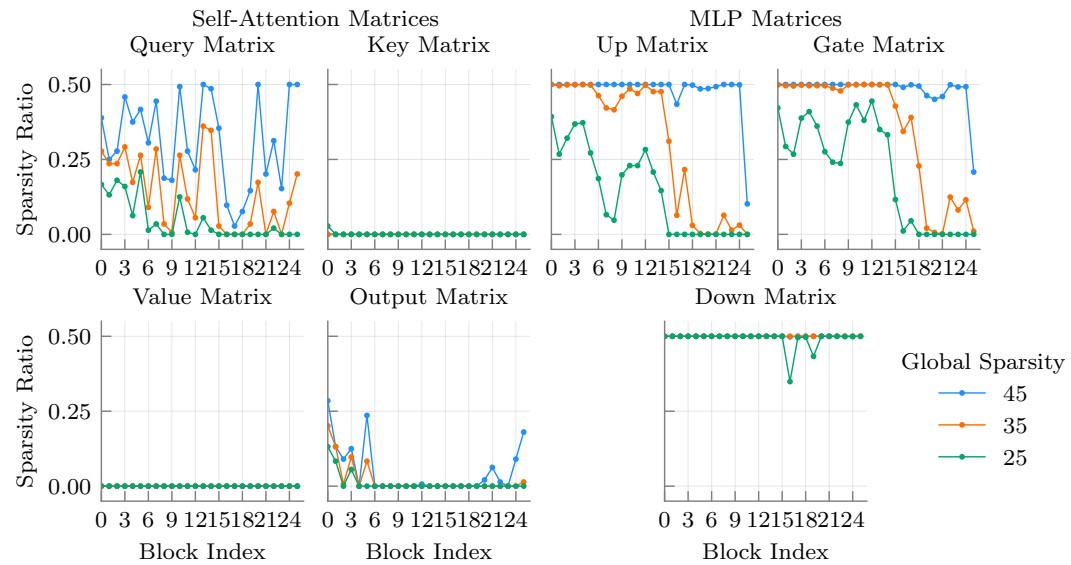

Figure 4: Sparsity distribution across Attention and MLP layers under varying global sparsity budgets in Gemma-3 1B.

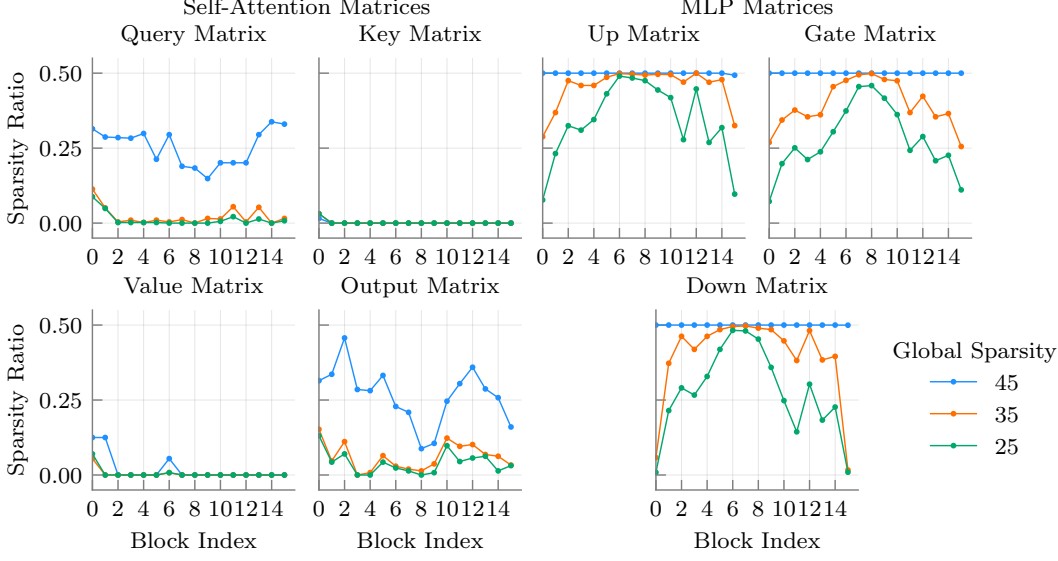

Figure 5: Sparsity distribution across Attention and MLP layers under varying global sparsity budgets in LLaMA-3.2 1B.

**Local pruning.** Local approaches prune each layer independently, typically by minimizing reconstruction error within that layer. A seminal example is Optimal Brain Surgeon (OBS) (Hassibi et al., 1993; Frantar & Alistarh, 2022), which leverages second-order information to identify and remove weights while updating the remaining parameters to compensate for loss. While highly principled, the quadratic cost of computing and inverting the Hessian makes OBS infeasible for large models.

Recent work adapts these ideas to LLM-scale pruning. SparseGPT (Frantar & Alistarh, 2023) formulates layer-wise pruning as a sparse regression problem, enabling efficient approximations of OBS that scale to billion-parameter models. Thanos (Ilin & Richtarik, 2025) further improves accuracy by employing multi-column approximations to reduce error accumulation. Wanda (Sun et al., 2023), on the other hand, discards explicit weight updates and instead uses a simple magnitude-

activation criterion with calibration data, yielding competitive quality with extremely fast runtimes. Despite their efficiency, local methods often suffer from limited capacity to recover accuracy since pruning decisions ignore cross-layer dependencies.

**Global pruning.** Global approaches aim to jointly optimize pruning decisions across layers, typically leading to better overall trade-offs. Optimal Brain Damage (OBD) (LeCun et al., 1989) is an early global method that estimates weight saliency using the diagonal Hessian. Extensions such as WoodFisher (Singh & Alistarh, 2020) approximate the Hessian via Kronecker factorizations, making computation more tractable but still challenging for modern LLMs (Mozaffari et al., 2023).

More recent approaches bypass costly second-order computations. MaskLLM (Fang et al., 2024) formulates pruning as a binary classification task (keep vs. prune) and solves it using standard optimizers such as AdamW (Loshchilov, 2017), achieving strong results even under hardware-friendly structured sparsity (e.g., 2:4). ProxSparse (Liu et al., 2025) instead adopts a proximal regularization framework, reducing the overhead of MaskLLM while trading off some pruning accuracy. These works highlight the tension between pruning quality and efficiency: global methods often achieve higher accuracy but remain more computationally expensive than simple one-shot local pruning.

### F.2 COMPLEMENTARY COMPRESSION TECHNIQUES

Beyond pruning, several orthogonal compression techniques are widely used and can be combined with sparsity for additional gains. *Quantization* reduces the bit precision of parameters and activations, e.g., from 32-bit floating point to 8- or 4-bit integers, thereby reducing memory footprint and accelerating inference (Gholami et al., 2022; Rokh et al., 2023).

*Low-rank adaptation* methods decompose weight matrices into smaller factors, effectively reducing parameter counts while maintaining expressivity. Recent approaches such as LQ-LoRA (Guo et al., 2023), SLiM (Mozaffari et al., 2025a), and SLoPe (Mozaffari et al., 2025b) demonstrate that low-rank structures can be used both for efficient fine-tuning and for direct model compression.

Finally, *knowledge distillation* (Gou et al., 2021) transfers knowledge from a large teacher model to a smaller student, yielding compact models that retain much of the teacher's performance. These methods are complementary to pruning, and hybrid frameworks that integrate sparsity, quantization, and low-rank factorization represent a promising direction for achieving high compression ratios without sacrificing accuracy.

## G COMPARISON WITH UNSTRUCTURED SPARSITY

In this section, we compare the quality of the models pruned with PATCH against other unstructured sparsity methods. Table 16 summarizes the average accuracy of the models across eight downstream tasks and the model perplexity on WikiText2 dataset. The results indicate that while unstructured sparsity consistently outperforms the hybrid sparsity, the gap between the two is not significant, showing that PATCH is helping bridging the gap between unstructured sparsity and semi-structured sparsity.

## H LAYER-WISE SPARSITY DISTRIBUTION COMPARISON WITH OTHER WORK

We compare the sparsity allocation learned by PATCH with OWL (Yin et al., 2025) and AlphaPruning (Lu et al., 2024). To ensure a robust baseline, we performed extensive hyperparameter sweeps for both methods and selected the configurations that achieved the best perplexity when applying Wanda as the underlying pruning operator:

- OWL: We swept $\lambda \in \{0.01, 0.03, 0.05, 0.08, 0.1\}$ and $M \in \{3, 5, 7, 10\}$.
- AlphaPruning: We swept 20 values of the temperature parameter $\tau$ between 0 and 0.5.

A critical distinction is that OWL and AlphaPruning are primarily designed for unstructured sparsity. While OWL includes an N:M structured variant, it is restricted to a fixed 50% global sparsity. Furthermore, OWL allocates sparsity at the block level (assigning uniform sparsity to all matrices

Table 16: Model quality (average accuracy across eight zero-shot tasks and perplexity on WikiText2 dataset) for PATCH, Wanda, and SparseGPT. For models with less than or equal to 1B parameters, PATCH$^{\text{Joint}}$ optimizes both dense tile locations and sparsity patterns, while for larger models PATCH$^{\text{Tile}}$ optimizes only dense tile locations with frozen sparsity patterns, both using Dense/2:4 Tiles pattern allowing continuous sparsity ratios and flexible tradeoffs between sparsity and model quality. Wanda and SparseGPT are unstructured pruning methods.

| Sparsity | Method | Pattern | Qwen-2.5 0.5B | | LLaMA-3.2 1B | | Gemma-3 1B | | LLaMA-2 7B | | LLaMA-3.1 8B | |
|---|---|---|---|---|---|---|---|---|---|---|---|---|
| | | | Acc (% ↑) | PPL (↓) | Acc (% ↑) | PPL (↓) | Acc (% ↑) | PPL (↓) | Acc (% ↑) | PPL (↓) | Acc (% ↑) | PPL (↓) |
| 45% | PATCH | Dense/2:4 Tiles | 40.29 | 14.57 | 42.08 | 12.23 | 42.80 | 11.96 | 48.99 | 6.55 | 53.60 | 8.20 |
| 45% | Wanda | Unstructured | 41.45 | 18.81 | 40.76 | 16.56 | 42.87 | 25.38 | 52.72 | 6.36 | 55.67 | 8.24 |
| 45% | SparseGPT | Unstructured | 42.31 | 17.65 | 42.66 | 15.01 | 43.52 | 22.26 | 52.77 | 6.46 | 56.70 | 8.21 |
| 35% | PATCH | Dense/2:4 Tiles | 41.15 | 13.84 | 42.72 | 11.67 | 43.30 | 11.48 | 50.08 | 6.18 | 55.28 | 7.89 |
| 35% | Wanda | Unstructured | 43.46 | 15.04 | 44.60 | 11.95 | 45.50 | 16.98 | 54.37 | 5.87 | 58.68 | 7.02 |
| 35% | SparseGPT | Unstructured | 44.66 | 14.79 | 45.62 | 11.68 | 45.45 | 16.92 | 54.18 | 5.92 | 58.81 | 7.07 |
| 25% | PATCH | Dense/2:4 Tiles | 42.39 | 13.47 | 43.81 | 11.00 | 44.07 | 11.17 | 51.58 | 5.86 | 56.48 | 7.34 |
| 25% | Wanda | Unstructured | 45.70 | 13.70 | 46.50 | 10.46 | 46.56 | 15.14 | 54.60 | 5.65 | 59.80 | 6.54 |
| 25% | SparseGPT | Unstructured | 45.28 | 13.63 | 46.52 | 10.42 | 46.37 | 15.05 | 54.71 | 5.68 | 59.52 | 6.55 |

within a Transformer block). Although the authors propose a weight-wise variant, they report, and our experiments confirm, that it yields inferior performance. In contrast, PATCH operates with tile-level granularity within a semi-structured constraint, offering a unique combination of hardware acceleration and fine-grained control.

As seen in Figures 6 and 7, both baselines produce relatively flat sparsity distributions across the model. AlphaPruning exhibits only minor fluctuations in the middle layers, whereas PATCH discovers distinct, highly non-uniform patterns (e.g., preserving Attention layers while aggressively pruning MLP blocks).

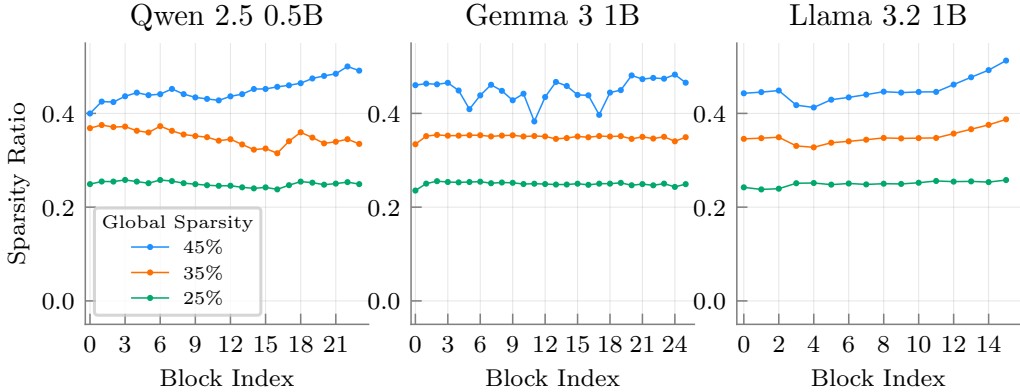

Figure 6: Layer-wise sparsity distribution of OWL across models and global sparsity budgets.

For OWL, we performed a sweep over the $\lambda$ parameter in $0.01, 0.03, 0.05, 0.08, 0.1$ and the $M$ parameter in $3, 5, 7, 10$ for each model. For AlphaPruning, we swept 20 values of $\tau \in [0, 0.5]$. For both methods, we selected the hyperparameters that achieved the best perplexity using Wanda as the pruning metric.

# I VARIATION ACROSS SEEDS

We evaluate PATCH on Qwen-2.5 0.5B and Llama-3.2 1B across different seeds. As shown in Table 17, model performance remains consistent across seeds.

In addition, we include the corresponding sparsity allocations across layers (Figure 9) and across individual weight matrices for both Llama-3.2 1B (Figure 10) and Qwen-2.5 0.5B (Figure 11). From these, we observe the following:

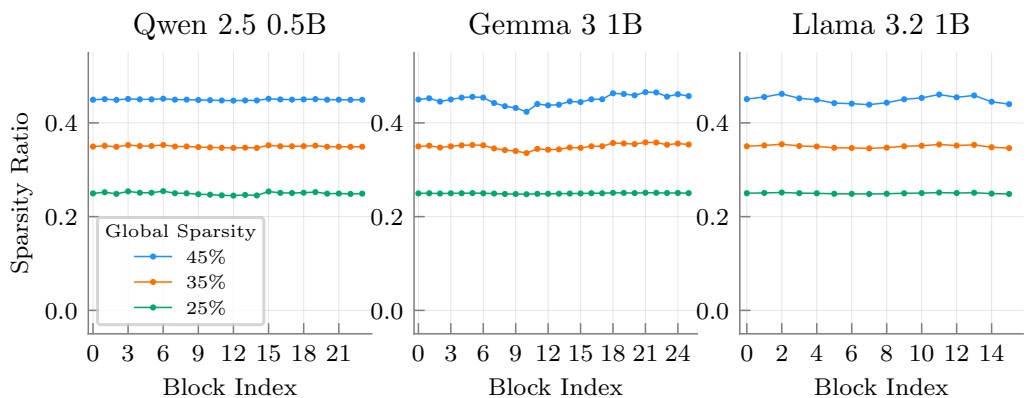

Figure 7: Layer-wise sparsity distribution of AlphaPruning across models and global sparsity budgets.

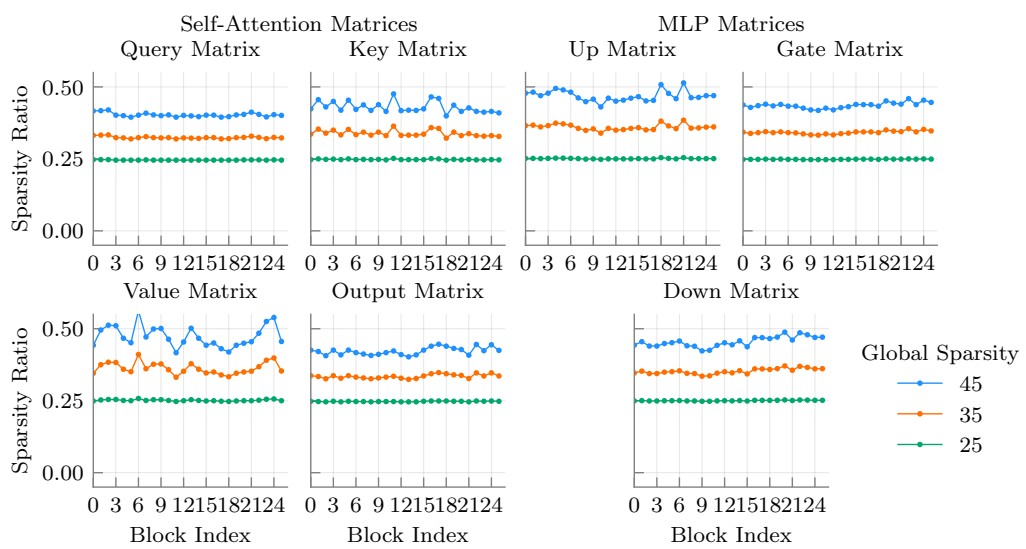

Figure 8: AlphaPruning sparsity distribution across attention and MLP layers under varying global sparsity budgets in Gemma-3 1B.

- At the global block level, the optimization consistently identifies the middle Transformer blocks as the most redundant (receiving the highest sparsity), while the initial and (especially) final blocks are pruned less.

- At the weight-matrix level, The allocation of sparsity between attention and MLP modules remains stable. For example, in Llama-3.2 1B (Figure 10), the Key and Value matrices consistently remain dense across all seeds, while the Query and MLP matrices absorb the majority of the sparsity

While the specific tile indices may vary slightly due to the stochastic sampling, the macroscopic pruning strategy learned by PATCH is highly reproducible and robust to initialization.

Table 17: Perplexity across seeds for Qwen-2.5 0.5B and Llama-3.2 1B.

| Model | Sparsity (%) | Seed 0 (default) | Seed 25 | Seed 26 | Seed 42 |
|---|---|---|---|---|---|
| Qwen-2.5 0.5B | 25 | 13.47 | 13.41 | 13.38 | 13.36 |
| | 35 | 13.84 | 13.89 | 13.85 | 13.84 |
| | 45 | 14.56 | 14.59 | 14.61 | 14.49 |
| Llama-3.2 1B | 25 | 11.00 | 11.09 | 11.03 | 11.21 |
| | 35 | 11.67 | 11.72 | 11.56 | 11.86 |
| | 45 | 12.23 | 12.32 | 12.26 | 12.55 |

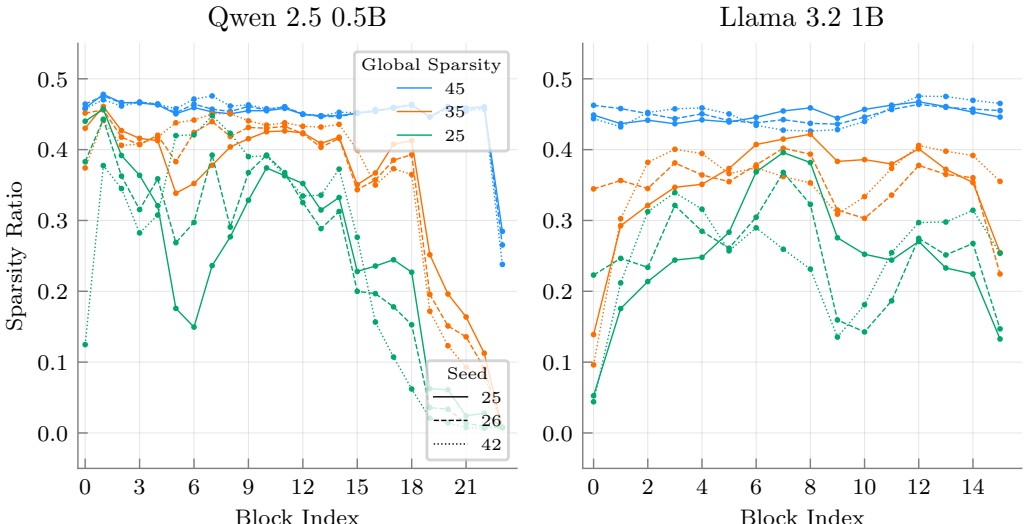

Figure 9: Layer-wise sparsity distribution of PATCH across seeds.

## J  FINE-TUNING AFTER MASK TRAINING

In this section, we present results from fine-tuning the remaining unpruned weights after mask training. We conducted a brief 5.6M-token run on the SlimPajama dataset so that the fine-tuning phase matches the mask search of PATCH.

Fine-tuning yields a consistent improvement in average zero-shot accuracy (e.g., +0.8% for Llama-3.2 1B at 25% sparsity and +0.6% for Qwen-2.5 0.5B at 45% sparsity), as seen on on Table 18. Interestingly, we observe a slight degradation in perplexity. We attribute this to the limited calibration data (5.6M tokens) compared to the trillions of tokens seen during pre-training; short fine-tuning can sometimes slightly drift the language modeling distribution while sharpening downstream task performance.

These results confirm that PATCH creates a high-quality sparsity topology that serves as a strong foundation. While the mask alone delivers state-of-the-art performance, subsequent fine-tuning, even with a limited budget, can further recover accuracy, offering a flexible path for users with additional compute resources.

## K  IMPACT OF SPARSE FINE-TUNING UNDER FIXED COMPUTE BUDGET

To assess whether the performance gains of PATCH are solely due to training, we conducted a controlled experiment comparing PATCH against fine-tuned (FT) one-shot baselines under a strictly fixed compute budget. We fine-tuned the weights of Wanda and SparseGPT (2:4 sparsity) models on

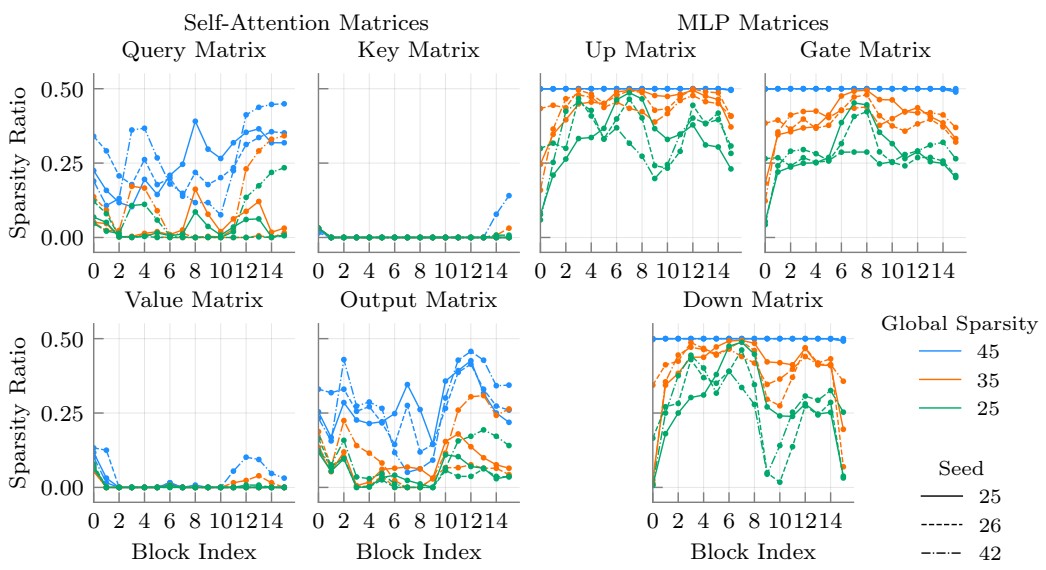

Figure 10: Sparsity distribution across attention and MLP layers under varying global sparsity budgets in Llama-3.2 1B across seeds.

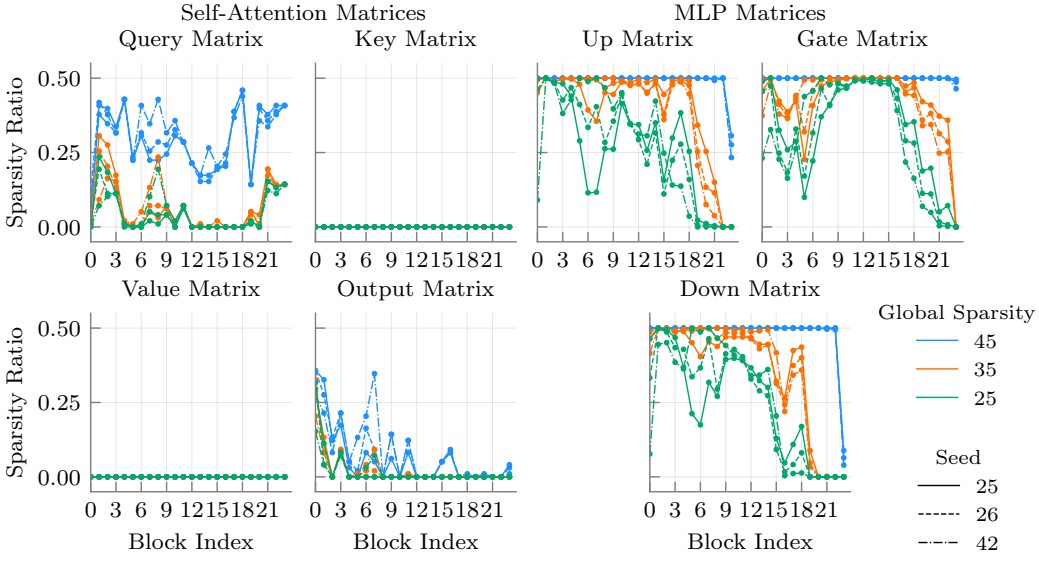

Figure 11: Sparsity distribution across attention and MLP layers under varying global sparsity budgets in Qwen-2.5 0.5B across seeds.

the SlimPajama dataset for 5.6M tokens. We trained the PATCH masks for the equivalent compute budget (matching the 5.6M token run).

As shown in Table 19, PATCH consistently outperforms the fine-tuned baselines, even when the baselines are allowed to update their weights. For example, on LLaMA-3.2 1B, PATCH at 45% sparsity achieves 42.08% accuracy, surpassing SparseGPT + FT (41.74%) and Wanda + FT (41.27%). This demonstrates that under a fixed compute budget, learning a flexible, non-uniform sparsity mask yields better performance than fine-tuning weights with a rigid, uniform mask.

Table 18: Model quality (average accuracy across eight zero-shot tasks and perplexity on WikiText2 dataset) for PATCH after a short fine-tuning.

| Model | Sparsity | Method | Wiki PPL ($\downarrow$) | Avg Acc (% $\uparrow$) |
|---|---|---|---|---|
| **Qwen-2.5 0.5B** | 45% | PATCH$^{\text{Joint}}$ | **14.56** | 40.29 |
| | | PATCH$^{\text{Joint}}$ + FT | 14.96 | **40.87** |
| | 35% | PATCH$^{\text{Joint}}$ | **13.84** | 41.15 |
| | | PATCH$^{\text{Joint}}$ + FT | 14.32 | **41.59** |
| | 25% | PATCH$^{\text{Joint}}$ | **13.47** | 42.39 |
| | | PATCH$^{\text{Joint}}$ + FT | 13.85 | **42.55** |
| **LLaMA-3.2 1B** | 35% | PATCH$^{\text{Joint}}$ | **11.67** | 42.72 |
| | | PATCH$^{\text{Joint}}$ + FT | 12.02 | **43.50** |
| | 25% | PATCH$^{\text{Joint}}$ | **11.00** | 43.81 |
| | | PATCH$^{\text{Joint}}$ + FT | 11.36 | **44.61** |

Table 19: Comparison of PATCH vs. Fine-Tuned (FT) Baselines under a Fixed Compute Budget.

| Model | Method | Sparsity (%) | PPL ($\downarrow$) | Avg Acc ($\uparrow$) |
|---|---|---|---|---|
| **Qwen-2.5 0.5B** | Wanda (2:4) | 50 | 72.48 | 32.97 |
| | Wanda + FT | 50 | 15.21 | 39.98 |
| | SparseGPT (2:4) | 50 | 36.59 | 34.81 |
| | SparseGPT + FT | 50 | 16.98 | 38.89 |
| | **PATCH$^{\text{Joint}}$** | **45** | **14.56** | **40.29** |
| | **PATCH$^{\text{Joint}}$** | **35** | **13.84** | **41.15** |
| | **PATCH$^{\text{Joint}}$** | **25** | **13.47** | **42.39** |
| **LLaMA-3.2 1B** | Wanda (2:4) | 50 | 78.18 | 31.61 |
| | Wanda + FT | 50 | 13.98 | 41.27 |
| | SparseGPT (2:4) | 50 | 32.73 | 35.55 |
| | SparseGPT + FT | 50 | 13.54 | 41.74 |
| | **PATCH$^{\text{Joint}}$** | **45** | **12.23** | **42.08** |
| | **PATCH$^{\text{Joint}}$** | **35** | **11.67** | **42.72** |
| | **PATCH$^{\text{Joint}}$** | **25** | **11.00** | **43.81** |

## L   LANGUAGE MODEL USAGE IN PAPER

We used language models to enhance the readability of the manuscript, correct grammatical and typographical errors, and ensure conformity with the ICLR author guidelines. Beyond their application in benchmark evaluations and experimental procedures, language models were not employed in any other aspect of this study.

## M   REPRODUCIBILITY STATEMENT

To support reproducibility, we release a repository linked in the abstract footnote that contains our implementation, training scripts, and evaluation pipeline. The paper outlines the method in § 3 and provides a thorough experimental description in § 5. Appendix D discusses the hyperparameter values in our work and additional information about our implementation. These materials collectively allow others to replicate our experiments and validate the claims made in the paper.

