# OpenReview forum: "PATCH: Learnable Tile-level Hybrid Sparsity for LLMs"
_ICLR.cc/2026/Conference — Submitted to ICLR 2026_

### Official Review · Reviewer_MP3s · 2025-10-27

**Soundness:** 2
**Presentation:** 3
**Contribution:** 2
**Rating:** 4
**Confidence:** 4

**Summary:**

This paper proposes a "hybrid sparse" framework PATCH. Instead of simply mixing unstructured and structured sparseness, it divides the weight matrix into "tiles" and dynamically decides whether to keep each tile dense or adopt hardware-friendly 2:4 sparseness through a learnable mask. The effectiveness of this method has been verified on models with parameters ranging from 0.5B to 8B. However, since no experiments are conducted on larger models (e.g., 70B), the generalization of the proposed method cannot be verified.

**Strengths:**

- The proposed "hybrid sparse" framework is not a simple mixture of unstructured and structured sparse.
- The paper is easy to follow.
- The ablation experiment is very thorough.

**Weaknesses:**

- This paper does not report the computational cost of the masking process, which will significantly affect the effectiveness of the pruning algorithm.
- No experiments are conducted on larger models (e.g., llama3-70B), which resulted in a lack of generalization of the algorithm.
- Can the proportion of weight pruning be increased to 80-90%? What is the performance of the proposed method in this case?
- Table 3 shows the impact of different tile sizes on the performance of the pruned model. So how do different tile sizes affect the performance acceleration of the pruned model?
- If the authors use other datasets to train the mask, will the effect be different?

**Questions:**

Compared with existing channel pruning and layer pruning methods, which one has better hardware acceleration effect?

---

> ### Author Response · Authors · 2025-11-22
>
> We thank the reviewer for their insightful feedback. In the following, we provide detailed answers to the questions and concerns raised by the reviewer.
>
> Formatting Convention: To ensure clarity, tables and figures introduced in this rebuttal are referred to simply by their number (e.g., Table 1). References to content in the original manuscript use the format `Content-Page-Number` (e.g., `Table 3-Page 7`).
>
> ---
>
> # Computational Cost of PATCH
> We thank the reviewer for raising this important point regarding the practicality of our method. While we included these details in `Appendix-D-Page-19`, we recognize they are central to the method's evaluation and summarize them here for clarity. Training is accelerated via data parallelism across a single node with 4 H100 GPUs. In this setup, PATCH-Joint​ requires 18 and 24 GPU-hours on the 0.5B and 1B models, respectively, while PATCH-Tile requires 84 and 96 GPU-hours on the 7B and 8B models.
>
> ---
>
> # Experiments on Larger Models
>
> We did not scale beyond 8B parameters strictly due to the high computational cost associated with training larger models end-to-end.
>
> While the 8B model could be trained on a single 80GB GPU, scaling to 70B parameters increases the computational requirements by an order of magnitude. For context, PATCH-Tile on an 8B model required approximately 96 H100 GPU-hours. Linearly extrapolating this to a 70B model (excluding the additional communication overhead of multi-node training) implies a cost of over 840+ H100 GPU-hours per run. This exceeds our available academic computational budget.
>
> However, regarding the expected behavior: **We anticipate that PATCH would perform even better at the 70B scale.** It is well-established in the literature (and observed in our own 0.5B vs. 8B comparisons) that larger models possess greater redundancy and are generally more robust to pruning. Since PATCH effectively exploits redundancy via learnable masking, applying it to a 70B model would likely yield higher accuracy retention and potentially allow for even more aggressive sparsity ratios than what is achievable on 7B/8B models.
>
> ---
>
> # Increasing Sparsity to 80-90%.
> The upper bound of sparsity in PATCH is determined by the hardware constraints we target. Since our framework is designed to leverage NVIDIA Tensor Cores, it exclusively utilizes the 2:4 semi-structured pattern. PATCH dynamically assigns one of two states to each tile: Dense (0% sparsity) or 2:4 Sparse (50% sparsity). Consequently, the global sparsity is a weighted average of these two states and is mathematically bounded between 0% and 50%. It is therefore structurally impossible to reach 80-90% sparsity under this specific hardware-aligned constraint.
>
> We did explore extending the search space to include a "fully pruned" option (100% sparsity), which would allow the global average to exceed 50%. However, in our preliminary experiments, allowing complete tile removal resulted in severe accuracy degradation. We found that maintaining at least the 2:4 structure in sparse regions was necessary to preserve model performance.
>
> To achieve 80-90% sparsity while retaining acceleration, one would require hardware support for higher-ratio N:M patterns (e.g., 1:4, which yields 75% sparsity) or block-sparse hardware support. As current generations of GPUs (Ampere/Hopper) generally optimize specifically for 2:4, we restricted our scope to this pattern to ensure that our reported sparsity translates into real-world speedups.

---

> > ### Author Response · Authors · 2025-11-22
> >
> > # Impact of Different Tile Size on Model Speedup
> >
> > In our implementation, the acceleration is determined by STOICC’s autotuning phase (similar to OpenAI Triton), which benchmarks a set of candidate tile sizes and selects the fastest kernel configuration for each layer. For this experiment, we restrict the maximum tile sizes to 64x64 or 32x64 (smaller dimensions are not supported by STOICC’s metadata layout and compression constraints). The results are in the following table :
> >
> >
> > | Sparsity (%) | Maximum Tile Size        | Throughput (tok/s)      | Speedup vs. dense |
> > |--------------|---------------------------|--------------------------|--------------------|
> > | 0            | —                         | 1023.80                  | 1x                 |
> > | 25           | (64,64)-(32,64)        | 1207.08-1075.98       | 1.18x-1.05x     |
> > | 35           | (64,64)-(32,64)        | 1288.54-1142.65       | 1.26x-1.12x     |
> > | 45           | (64,64)-(32,64)        | 1382.13-1166.19       | 1.35x-1.14x     |
> >
> > * Table 1 : Throughput of LLaMA-2 7B with mixed sparsity compared to the dense model. Measurements taken on an A6000 GPU with batch size 16, sequence length 128, tokens generated 128. Throughput is reported in tokens processed/sec.*
> >
> > Since STOICC already mostly prefers 64×64 tiles in our main experiments, limiting the autotuner to 64×64 has only a small effect on throughput. In contrast, forcing it to use only 32×64 tiles results in slower kernels, as this tile is too small to fully benefit from tensor-core acceleration.
> >
> > As shown in `Table-3-Page-7` of the paper, varying the tile size (e.g., 64 vs 128) has negligible impact on accuracy. Therefore, in our main experiments, we define the problem using larger 128×128 tiles. This provides the compiler’s autotuner with the maximum flexibility to subdivide the matrix and select the optimal kernel configuration (typically 64×64) for execution.
> >
> > Finally, we note that these throughput variations reflect system-level kernel behavior in STOICC/Triton rather than properties of PATCH itself. PATCH is a pruning method, and we simply use STOICC as the backend to demonstrate that the resulting structured masks can be accelerated. The specifics of how different tile sizes interact with GPU kernels are an implementation detail of the compiler, not a component of our work.
> >
> > ---
> >
> > # Mask Training on Different Datasets
> >
> > Thank you for this suggestion. We applied PATCH to Qwen-2.5 0.5B and Llama-3.2 1B using the same training setup, but with the mask trained on the **C4 dataset instead of SlimPajama**.
> >
> > As shown in the table below, while we observe a marginal increase in perplexity when training on C4 (expected due to the domain shift), the downstream average accuracy remains stable (within ~0.6% of the SlimPajama baseline).
> >
> >
> >
> >
> > | Model          | Sparsity / Method | C4 PPL | C4 Avg (%) | SlimPajama PPL | SlimPajama Avg (%) |
> > |----------------|-------------------|--------|-------------|-----------------|---------------------|
> > | Qwen 2.5 0.5B  |        |   |        |            |                |
> > |   | MaskLLM 50%       | 16.27  | 38.78       | 15.22           | 39.33               |
> > |   | Ours 45%          | 15.30  | 40.02       | 14.56           | 40.29               |
> > |   | Ours 35%          | 14.34  | 40.94       | 13.84           | 41.15               |
> > |   | Ours 25%          | 13.81  | 41.91       | 13.47           | 42.39               |
> > | Llama 3.2 1B |        |   |        |            |                |
> > |  | MaskLLM 50%       | 14.23  | 40.97       | 12.93           | 41.04               |
> > |  | Ours 45%          | 13.10  | 41.73       | 12.23           | 42.08               |
> > |  | Ours 35%          | 12.07  | 42.98       | 11.67           | 42.72               |
> > |  | Ours 25%          | 11.42  | 43.37       | 11.00           | 43.81               |
> > Table 2: Comparison of PATCH training with SlimPajama and C4 dataset
> >
> >
> > The results are consistent across datasets, and PATCH remains stronger than the baselines in all cases.

---

> > > ### Comment · Reviewer_MP3s · 2025-11-27
> > >
> > > Thank you for your detailed reply, but the effectiveness on large models (e.g., 70B) remains my biggest concern.

---

> > > > ### Author Response · Authors · 2025-11-28
> > > >
> > > > We thank the reviewer for their feedback. We understand that validating performance on 70B scale models is a significant concern. We would like to address this by clarifying the computational constraints and the scientific context of our evaluation.
> > > >
> > > > 1. **Consistency with Accepted State-of-the-Art Baselines**: The standard for rigorous evaluation in recent pruning literature typically caps at the 7B–13B range due to high computational barriers. For example, MaskLLM [1] (NeurIPS 2024) and ProxSparse [2] (ICML 2025) both limit their experimental validation to models up to 13B parameters. Our evaluation on the 8B model is consistent with these accepted benchmarks. Requiring 70B experiments would impose a resource burden significantly higher than that of comparable accepted works in the field.
> > > >
> > > > 2. **Computational Infeasibility for Academic Research**: Based on our 8B training time, linear extrapolation suggests that training the mask for the 70B model would require approximately 850+ H100 GPU-hours (roughly 9 days of continuous training on our 4-GPU node, assuming linear scaling without memory bottlenecks). The cost is likely higher since the model doesn't fit on a single GPU (it requires about 140 GB just to store the weights), which forces the use of pipeline or tensor parallelism and significantly slows down throughput. This scale of experimentation exceeds the capacity of our academic laboratory.
> > > >
> > > > 3. **Scientific Validity via Scaling Laws**: From a scientific perspective, results on smaller models are generally considered a "lower bound" for pruning performance. It is well-established in the literature (e.g., SparseGPT [3]; Wanda [4]) that smaller models are significantly harder to prune than larger ones due to lower parameter redundancy. We have demonstrated that PATCH is effective on the most difficult cases (0.5B and 1B models) and remains robust as we scale to 7B and 8B. Since 70B models possess even greater redundancy, there are strong theoretical grounds to expect that PATCH would perform equal to or better than the reported results at scale.
> > > >
> > > > 4. **Generalization Across Architectures**: Finally, we have verified PATCH not just on one model family, but across diverse architectures including LLaMA-2, LLaMA-3, and Qwen, consistently outperforming baselines. This confirms that our "hybrid sparse" tile formulation generalizes well across different transformer implementations and is not overfitting to a specific model size or structure.
> > > >
> > > > We hope this clarifies that the omission of 70B results is due to strict hardware limitations common in academic research, rather than a limitation of the method's potential.
> > > >
> > > > ---
> > > >
> > > > [1] Fang et al., MaskLLM: Learnable Semi-Structured Sparsity for Large Language Models, NeurIPS 2024
> > > >
> > > > [2] Liu et al., ProxSparse: Regularized Learning of Semi-Structured Sparsity Masks for Pretrained LLMs, ICML 2025
> > > >
> > > > [3] Frantar et al., SparseGPT: Massive Language Models Can Be Accurately Pruned in One-Shot, ICML 2023
> > > >
> > > > [4] Sun et al., A Simple and Effective Pruning Approach for Large Language Models, ICLR 2024

---

### Official Review · Reviewer_Vov1 · 2025-10-31

**Soundness:** 2
**Presentation:** 3
**Contribution:** 2
**Rating:** 2
**Confidence:** 5

**Summary:**

This paper proposes to learn a heterogeneous sparsity for weight tensors which combines dense and 2:4 sparse tiles. The introduction of dense tiles enables the model sparsity to range from 0-50% and vary between layers. Gumbel-Softmax reparameterization is used to learn soft masks during training which converge to hard masks by the end of training as the distribution hyperparameters, $\tau$ & $k$, are modified accordingly. The proposed method, PATCH, at 45% sparsity exceeds the accuracy of 2:4 one-shot pruning methods and MaskLLM on loglikelihood-based question answering language benchmarks. The method is benchmarked using Triton kernels from the STOICC library and demonstrates up to 1.38x acceleration at 45% sparsity on A6000 GPUs.

**Strengths:**

* Heterogeneous sparsity within a single weight tensor is an under-explored setting for exploiting sparsity for efficiency gains. The proposed method is hardware friendly for commodity hardware.
* The layer-wise sparsity distribution is learned rather than imposed via heuristic. The paper includes some intriguing details on which layers are generally more sparse, such as the FFN layers, whereas the key/value projections remain more dense.
* A number of ablations are provided, including: joint mask and tile vs. tile-only search, tile size, layerwise sparsity allocation.
* The observed acceleration on A6000s at 45% acceleration approaches the expected benefit of pure 2:4 sparsity.

**Weaknesses:**

Overall, my main concerns relate to gauging the accuracy and throughput of PATCH in comparison with simpler approaches.

# Major concerns
The following represent key weaknesses that must be addressed to increase the rating:
* Heterogeneous intra vs. inter layer sparsity: The sparsity distributions suggest that PATCH predominantly prunes the FFN layers. A robust baseline may be to simply prune only the FFN linear layers to 2:4 sparsity. For instance, with Llama-3.1-8B-Instruct, excluding layernorm parameters, ~81% of the parameters associated with each decoder block are contained in the FFN module. Pruning just the FFN modules to 2:4 will yield a global sparsity of 40%. For direct comparison with PATCH at 35%, 4 decoder layers can be left dense to yield a global sparsity of 35.3%, for instance, the last 4 decoder blocks.
* Sparse fine-tuning: PATCH requires significant training resources to learn the optimal patch assignments and specific 2:4 masks. Given a fixed compute budget, simply fine-tuning the sparse weights from a one-shot pruning method may exceed both the accuracy and throughput of PATCH.
* 2:4 benchmarks: What is the throughput of global 2:4 pruned models under the same settings that PATCH was benchmarked? What is the throughput when only the FFN layers are pruned to 2:4?
* Sparsity-matched accuracy comparisons: Direct comparison of 45% PATCH and 2:4 sparse models from other compression methods is challenging. How do select baselines such as SparseGPT pruned to 3:4 compare with the 25% sparse PATCH models?

# Minor concerns
The following are minor concerns, typos, etc. which would improve the work but do not affect the final rating:
* Lack of generative evaluations: Loglikelihood-based QA benchmarks and PPL can be misleading when evaluating compressed models. Including a benchmark that assesses the generative outputs of the model, such as EvalPlus, would help establish how practical PATCH is for real-world tasks.
* Hopper benchmarks: Generally the efficiency gains from 2:4 sparsity have been less impressive on Hopper. It would be interesting to see benchmarks on the current generation of NVIDIA GPUs to assess the benefits of PATCH.
* L092: “while freezing the initial 2:4 mask fixed” -> “while freezing the initial 2:4 mask”.
* Section 5.3 quantization section: This paragraph is lacking details. Including further discussion on the compatibility of quantization with PATCH would be beneficial.
* Layer-wise sparsity distribution comparisons: Prior methods for determining optimal layer-wise sparsity ratios have been proposed. Comparisons between the distribution learned by PATCH and [1-3] may be interesting.

[1] L. Li et al., “Discovering Sparsity Allocation for Layer-wise Pruning of Large Language Models,” presented at the The Thirty-eighth Annual Conference on Neural Information Processing Systems, Nov. 2024. Available: https://openreview.net/forum?id=rgtrYVC9n4

[2] H. Lu, Y. Zhou, S. Liu, Z. Wang, M. W. Mahoney, and Y. Yang, “AlphaPruning: Using Heavy-Tailed Self Regularization Theory for Improved Layer-wise Pruning of Large Language Models,” Oct. 14, 2024, arXiv: arXiv:2410.10912. doi: 10.48550/arXiv.2410.10912.

[3] L. Yin et al., “Outlier Weighed Layerwise Sparsity (OWL): A Missing Secret Sauce for Pruning LLMs to High Sparsity,” Oct. 08, 2023, arXiv: arXiv:2310.05175. doi: 10.48550/arXiv.2310.05175.

**Questions:**

See major concerns in the Weaknesses section above.

---

> ### Author Response · Authors · 2025-11-22
>
> We thank the reviewer for their insightful feedback. In the following, we provide detailed answers to the questions and concerns raised by the reviewer.
>
> Formatting Convention: To ensure clarity, tables and figures introduced in this rebuttal are referred to simply by their number (e.g., Table 1). References to content in the original manuscript use the format `Content-Page-Number` (e.g., `Table 3-Page 7`).
>
> ---
>
> # Pruning the FFN Modules as a Baseline
>
> We implemented the suggested baseline by pruning only the FFN modules to 2:4 sparsity while keeping other layers dense. To achieve the specific global sparsity targets (e.g., 35%), we followed your recommendation to keep the final decoder blocks dense (specifically, the last 3–5 blocks depending on the model depth) and applied SparseGPT to the remaining FFN layers.
> As summarized below in Table 1, PATCH consistently outperforms this FFN-only heuristic across all models and sparsity levels.
>
> **Superior Accuracy:** PATCH achieves significantly lower perplexity and higher average accuracy than the baseline. For example, on LLaMA-2 7B at 45% sparsity, PATCH improves accuracy by over 2% compared to the baseline while maintaining lower perplexity (6.55 vs. 8.36).
>
> **Validation of Learnability:** While the reviewer correctly notes that FFN layers absorb the majority of sparsity (a trend confirmed by our Figure 2), these results demonstrate that a rigid heuristic (pruning all FFNs) is suboptimal. PATCH’s ability to learn flexible, non-uniform tile assignments (preserving critical FFN tiles and pruning redundant Attention tiles where appropriate) yields superior preservation of model knowledge.
>
>
> | Model            | Method                        | Sparsity (%) | PPL     | Avg    |
> |------------------|-------------------------------|--------------|---------|--------|
> | **Qwen-2.5 0.5B** |                               |              |         |        |
> |                  | PATCH                          | 45           | **14.57**   | **40.29**  |
> |                  | SparseGPT (FFN only)           | 44           | 24.79   | 36.84  |
> |                  | PATCH                          | 35           | **13.84**   | **41.15**  |
> |                  | SparseGPT (FFN except last 5)   | 35           | 20.46   | 38.33  |
> | **Llama 3.2 1B**  |                               |              |         |        |
> |                  | PATCH                          | 45           | **12.23**   | **42.08**  |
> |                  | SparseGPT (FFN only)           | 41           | 19.97   | 38.73  |
> |                  | PATCH                          | 35           | **11.67**   | **42.72**  |
> |                  | SparseGPT (FFN except last 3)   | 34           | 17.62   | 39.98  |
> | **Gemma 3 1B**    |                               |              |         |        |
> |                  | PATCH                          | 45           | **11.96**   | **42.80**  |
> |                  | SparseGPT (FFN only)           | 45           | 32.2    | 39.58  |
> |                  | PATCH                          | 35           | **11.48**   | **43.30**  |
> |                  | SparseGPT (FFN except last 3)   | 36           | 26.94   | 39.98  |
> | **Llama 2 7B**    |                               |              |         |        |
> |                  | PATCH                          | 45           | **6.55**    | **48.99**  |
> |                  | SparseGPT (FFN)                | 33           | 8.36    | 49.01  |
> |                  | PATCH                          | 35           | **6.18**    | **50.08**  |
> | **Llama 3.1 8B**  |                               |              |         |        |
> |                  | PATCH                          | 45           | **8.2**     | **53.60**  |
> |                  | SparseGPT (FFN only)           | 40           | 11.37   | 51.04  |
> |                  | PATCH                          | 35           | **7.89**    | **55.28**  |
> |                  | SparseGPT (FFN except last 4)   | 35           | 10.3    | 51.49  |
> Table 1: Comparison of PATCH Performance vs. 2:4 FFN-Only Pruning Across Sparsities

---

> > ### Author Response · Authors · 2025-11-22
> >
> > # Throughput Comparison: PATCH vs. FFN-Only Pruning
> >
> > We benchmarked the throughput of the FFN-only baseline under the same settings as PATCH (STOICC backend, A6000 GPU). As shown below in Table 2, the throughput of FFN-only pruning is nearly identical to that of PATCH at matched sparsity levels. However, as demonstrated in Table 1, PATCH achieves significantly better accuracy at these same speeds. By learning a flexible sparsity allocation, rather than heuristically pruning all FFN layers, PATCH maintains the same high throughput while preserving critical model weights. This makes PATCH a more efficient solution, offering a superior trade-off between performance and speed.
> >
> >
> > | Model        | Method | Sparsity (%) | Backend    | Throughput (tok/s) | Speedup |
> > |--------------|----------------|----------|------------|------------|---------|
> > | Llama-2 7B   |                |          |            |            |         |
> > |              | Dense          | 0        | cuBLAS     | 1023.80    | 1.00x |
> > |              | Patch          | 25       | STOICC     | 1212.79    | 1.18x |
> > |              | Patch          | 35       | STOICC     | 1304.46    | 1.27x |
> > |              | Patch          | 45       | STOICC     | 1410.20    | 1.38x |
> > |              | SparseGPT (FFN only)      | 33         | cuSPARSELt | 1276.67 | 1.25x |
> > |              | SparseGPT (FFN only)      | 33         | STOICC     | 1302.27 | 1.27x |
> > |              | SparseGPT      | 50       | cuSPARSELt | 1434.25    | 1.40x |
> > |              | SparseGPT      | 50       | STOICC     | 1501.91    | 1.47x |
> > | Llama-3.1 8B |                |          |            |            |         |
> > |              | Dense          | 0     | cuBLAS     | 908.02 | 1.00x |
> > |              | Patch          | 25     | STOICC     | 1027.44 | 1.13x |
> > |              | Patch          | 35     | STOICC     | 1079.02 | 1.19x |
> > |              | Patch          | 45     | STOICC     | 1164.57 | 1.28x |
> > |              | SparseGPT (FFN only except last 4)      | 35      | cuSPARSELt  | 1086.91 | 1.20x |
> > |              | SparseGPT (FFN only except last 4)      | 35      | STOICC  | 1093.27 | 1.20x |
> > |              | SparseGPT (FFN only)      | 40      | cuSPARSELt  | 1122.31 | 1.24x |
> > |              | SparseGPT (FFN only)      | 40     | STOICC | 1130.6 | 1.25x |
> > |              | SparseGPT      | 50     | cuSPARSELt  | 1176.45 | 1.30x |
> > |              | SparseGPT      | 50     | STOICC  | 1226.36 | 1.35x |
> >
> >
> > Table 2: Throughput and Speedup Comparison of PATCH vs. 2:4 (FFN Only) Pruned Models
> >
> > ---
> >
> > # Sparsity Matched Accuracy Comparison : 3:4 vs. PATCH 25%
> >
> >
> > We compared PATCH at 25% sparsity against SparseGPT pruned to a 3:4 pattern (keeping 3 out of 4 weights), as both result in 75% global density. As shown in Table 3, SparseGPT 3:4 achieves slightly better perplexity due to its higher flexibility (removing any 1 weight per 4 elements). However, this comes at the cost of acceleration.
> >
> > - **No Speedup for 3:4**: The 3:4 pattern is incompatible with NVIDIA Sparse Tensor Cores, which strictly require a 2:4 pattern (maximum 2 non-zeros per block of 4). Consequently, SparseGPT 3:4 models must run at dense speeds (1.0x).
> > - **Speedup for PATCH**: In contrast, PATCH 25% mixes dense tiles with hardware-friendly 2:4 tiles. This allows it to unlock real hardware acceleration (approx. 1.18x speedup) on standard GPUs.
> >
> > Additionally, in  `Appendix-G-Page-21`, we provided a comparison with unstructured sparsity at 45%, 35%, and 25%. While unstructured sparsity consistently outperforms the hybrid sparsity of PATCH, the performance gap is not significant. These results suggest that PATCH effectively bridges the performance gap between unstructured and semi-structured sparsity, offering a viable trade-off between accuracy and computational efficiency.
> >
> > | Model            | Method            | Sparsity (%) | PPL   | Avg (%) |
> > |------------------|-------------------|--------------|-------|---------|
> > | **Qwen-2.5 0.5B** | SparseGPT 3:4     | 25           | 13.99 | 43.34   |
> > |                  | PATCH             | 25           | 13.47 | 42.39   |
> > | **Llama 3.2 1B**  | SparseGPT 3:4     | 25           | 10.88 | 44.66   |
> > |                  | PATCH             | 25           | 11.00 | 43.81   |
> > | **Gemma 3 1B**    | SparseGPT 3:4     | 25           | 13.96  | 45.27   |
> > |                  | PATCH             | 25           | 11.17 | 44.07   |
> > | **Llama 2 7B**    | SparseGPT 3:4     | 25           | 5.55  | 54.08   |
> > |                  | PATCH             | 25           | 5.86  | 51.58   |
> > | **Llama 3.1 8B**  | SparseGPT 3:4     | 25           | 6.7 | 58.14   |
> > |                  | PATCH             | 25           | 7.34   | 56.48   |
> > Table 3 : Performance comparison of PATCH 25% and SparseGPT 3:4 Pruning

---

> ### Author Response · Authors · 2025-11-22
>
> # EvalPlus Benchmark
>
> Thank you so much for the insightful advice. We focused on log-likelihood-based benchmarks (such as perplexity and zero-shot accuracy) in our paper as they are standard metrics that enable direct comparison with prior art such as SparseGPT, Wanda, and MaskLLM. We fully agree that assessing generative outputs is essential for establishing practical utility. We are currently working on evaluating PATCH using EvalPlus as suggested by the reviewer, and will provide these additional generative results in the revised appendix.
>
> ---
>
> # Hopper Benchmarks
>
> We acknowledge the importance of evaluating on the latest hardware. However, the STOICC compiler, which provides the necessary backend support for our heterogeneous tile-level execution, does not currently support the NVIDIA Hopper architecture. Consequently, we were restricted to benchmarking on Ampere-generation GPUs (A6000, A100).
>
> Our reported speedups serve to demonstrate the feasibility of accelerating mixed dense and 2:4 sparse tiles using specialized kernels. This confirms that the PATCH format is hardware-friendly and can translate theoretical sparsity into real-world speedups on supported backends. We anticipate that as compiler support for Hopper matures, PATCH will similarly benefit from the H100's advanced Tensor Core capabilities.
>
> ---
>
> # Typos
>
> Thank you for pointing our typo in L092 out. We have made the changes in the main script.
>
> ---
>
> # Compatibility of Quantization with PATCH
>
> We appreciate the suggestion to elaborate on this critical aspect. We have significantly expanded Section 5.3 to explicitly discuss the orthogonality between PATCH and quantization. Specifically, we clarify that PATCH optimizes the sparsity pattern (selecting dense vs. 2:4 tiles) independent of the weight precision, allowing it to be seamlessly integrated with standard weight-only quantization techniques (e.g., 4-bit integers) without conflict.
>
> To empirically demonstrate this, we have revised `Table-6-Page-9` to more clearly showcase the following configurations:
> - MaskLLM (Int4): 2:4 sparsity with 4-bit quantization.
> - PATCH (Int4): Our mixed tile sparsity with 4-bit quantization.
> - PATCH + LoRA (Int4): Demonstrating compatibility with both quantization and low-rank recovery.
>
> The updated results illustrate that PATCH integrates robustly with 4-bit quantization, consistently outperforming the fully 2:4 quantized baselines while maintaining the computational benefits of the sparse format.
>
> ---
>
> # Sparsity Distribution Comparison with Prior Work
>
> To address this, we have conducted new experiments comparing PATCH against two state-of-the-art sparsity allocation methods: OWL [1] and AlphaPruning [2]. The resulting sparsity distributions are visualized in the newly added `Appendix-H-Page-22,23` (Figures 6, 7, and 8).
>
> To ensure a robust baseline, we performed extensive hyperparameter sweeps for both methods and selected the configurations that achieved the best perplexity when applying Wanda as the underlying pruning operator:
>
> - OWL: We swept $\lambda \in \{0.01, 0.03, 0.05, 0.08, 0.1\}$ and $M \in \{3, 5, 7, 10\}$.
> - AlphaPruning: We swept 20 values of the temperature parameter $\tau$ between 0 and 0.5.
>
> A critical distinction is that OWL and AlphaPruning are primarily designed for unstructured sparsity. While OWL includes an N:M structured variant, it is restricted to a fixed 50% global sparsity. Furthermore, OWL allocates sparsity at the block level (assigning uniform sparsity to all matrices within a Transformer block). Although the authors propose a weight-wise variant, they report, and our experiments confirm, that it yields inferior performance. In contrast, PATCH operates with tile-level granularity within a semi-structured constraint, offering a unique combination of hardware acceleration and fine-grained control.
>
> As illustrated in `Appendix-H-Page-21`, both baselines produce relatively flat sparsity distributions across the model. AlphaPruning exhibits only minor fluctuations in the middle layers, whereas PATCH discovers distinct, highly non-uniform patterns (e.g., preserving Attention layers while aggressively pruning MLP blocks).
> We were unable to include DSA as a baseline, as the available codebase is incomplete and the provided sparsity computation yields NaN outputs.
>
> [1] L. Yin et al., “Outlier Weighed Layerwise Sparsity (OWL): A Missing Secret Sauce for Pruning LLMs to High Sparsity,” Oct. 08, 2023, arXiv: arXiv:2310.05175. doi: 10.48550/arXiv.2310.05175.
>
> [2] H. Lu, Y. Zhou, S. Liu, Z. Wang, M. W. Mahoney, and Y. Yang, “AlphaPruning: Using Heavy-Tailed Self Regularization Theory for Improved Layer-wise Pruning of Large Language Models,” Oct. 14, 2024, arXiv: arXiv:2410.10912. doi: 10.48550/arXiv.2410.10912.

---

> > ### Author Response · Authors · 2025-11-22
> >
> > # Sparse Fine-Tuning and Compute Budget
> >
> > To address this concern, we conducted a controlled experiment comparing PATCH against fine-tuned (FT) one-shot baselines under a strictly fixed compute budget.
> >
> > - Baselines: We took models pruned via Wanda and SparseGPT (2:4 sparsity) and fine-tuned their weights on the SlimPajama dataset for 5.6M tokens using the AdamW optimizer (tuning learning rates between (5e-4, 1e-4, 5e-5, 1e-5)).
> > - PATCH: We trained the PATCH masks for the equivalent compute budget (matches the 5.6M token run). Note that standard PATCH learns the mask while keeping weights frozen.
> >
> > As shown in Table 3, PATCH consistently outperforms the fine-tuned baselines, even when the baselines are allowed to update their weights. For example, on Llama 3.2 1B, PATCH at 45% sparsity achieves 42.08% accuracy, surpassing SparseGPT + FT (41.74%) and Wanda + FT (41.27%).
> >
> > This leads to a compelling conclusion: under a fixed compute budget, learning a flexible, non-uniform sparsity mask (PATCH) yields better performance than fine-tuning weights with a rigid, uniform mask.
> >
> >
> > | **Model** | **Method** | **Sparsity** | **PPL ($\downarrow$)** | **Avg Acc ($\uparrow$)** |
> > | :--- | :--- | :--- | :--- | :--- |
> > | **Qwen-2.5 0.5B** | Wanda (2:4) | 50% | 72.48 | 32.97 |
> > | | Wanda + FT | 50% | 15.21 | 39.98 |
> > | | SparseGPT (2:4) | 50% | 36.59 | 34.81 |
> > | | SparseGPT + FT | 50% | 16.98 | 38.89 |
> > | | **PATCH Joint** | **45%** | **14.56** | **40.29** |
> > | | **PATCH Joint** | **35%** | **13.84** | **41.15** |
> > | | **PATCH Joint** | **25%** | **13.47** | **42.39** |
> > | **Llama 3.2 1B** | Wanda (2:4) | 50% | 78.18 | 31.61 |
> > | | Wanda + FT | 50% | 13.98 | 41.27 |
> > | | SparseGPT (2:4) | 50% | 32.73 | 35.55 |
> > | | SparseGPT + FT | 50% | 13.54 | 41.74 |
> > | | **PATCH Joint** | **45%** | **12.23** | **42.08** |
> > | | **PATCH Joint** | **35%** | **11.67** | **42.72** |
> > | | **PATCH Joint** | **25%** | **11.00** | **43.81** |
> > Table 3: Comparison of PATCH vs. Fine-Tuned (FT) Baselines under Fixed Compute Budget
> >
> > Note: While we restricted this comparison to mask-learning only for fairness, we also observed that performing additional weight fine-tuning on top of PATCH yields further gains, e.g., boosting Qwen-2.5 45% accuracy to 40.87%, widening the gap even further.

---

> > > ### Comment · Reviewer_Vov1 · 2025-11-27
> > >
> > > I thank the authors for the detailed rebuttal and clarifications. Based on the new experimental results and revised paper, I have increased my score from 2 to 6.

---

> > > > ### Author Response · Authors · 2025-11-28
> > > > **Thank you for the re-evaluation and score increase.**
> > > >
> > > > We sincerely thank the reviewer for the re-evaluation and the increased score. We are delighted that our rebuttal and the additional experiments addressed your concerns. We will ensure that all the new results and discussions are incorporated into the final version of the paper.
> > > >
> > > > Best,
> > > >
> > > > The Authors

---

### Official Review · Reviewer_bYHv · 2025-10-31

**Soundness:** 3
**Presentation:** 3
**Contribution:** 3
**Rating:** 6
**Confidence:** 3

**Summary:**

This work aims at solving the efficiency challenge of deploying LLMs by introducing PATCH, a new hybrid pruning framework that mixes dense weights and structured sparse weights at a tile level within each weight matrix. Vanilla pruning methods either use unstructured sparsity which is not hardware-efficient, or semi-structured 2:4 sparsity which is hardware-friendly but often hurts accuracy. PATCH bridges this gap by allowing a continuous sparsity ratio of 0-50% across the model. It partitions weight matrix into small tiles and uses a learnable binary tile mask (trained with Gumbel-Softmax relaxation to allow differentiable sampling) to assign each tile with either dense or 2:4 sparse, allowing important regions to remain fully dense while still exploiting 2:4 sparsity elsewhere. The authors also introduce a further memory-efficient variant of PATCH. This variant further reduces the number of learnable parameters by fixing the fine-grained 2:4 mask (e.g., initialized by a one-shot method like MaskLLM or SparseGPT) and only learning the tile assignments.

**Strengths:**

1. This work addresses a critical quality v.s. hardware efficiency tradeoff with hybrid apporach. By allowing a mixture of dense and 2:4-sparse tiles within weight matrices, PATCH achieves pruning ratios that were previously impossible with hardware-friendly methods.
2. PATCH shows SOTA results in terms of the aforementioned trade-off for pruned LLMs. Experiment results consistently show that PATCH-pruned models have higher accuracy / lower perplexity than those pruned by other methods at comparable sparsity level. Also this work demonstrates practical efficiency gains with up to 1.38x end-to-end speedup. Many prior pruning works struggled to translate sparsity into actual speed due to irregular patterns. PATCH overcomes that by construction (using a hardware-supported pattern and grouping it in tiles). Showing a non-uniform sparsity method yielding clock speed improvements is a commendable achievement.
3. Evaluation coverage is thorough -- multiple LLM model families (Llama, Qwen, Gemma) across model sizes from 500M up to 8B params on 8 diverse and repersentative zero-shot tasks (MMLU, PIQA, ARC-Easy, ARC-Challenge, WinoGrande, OpenBookQA, RACE, HellaSwag) plus WikiText2 perplexity
4. Using the Gumbel-Softmax for differentiable mask is a smart choice that leverages known methods in a new context

**Weaknesses:**

1. PATCH’s benefits rely on specialized support with native 2:4 sparse op (this is not a weakness of this work per-se but inherited from 2:4 sparsity for a sparsity-accuracy tradeoff). It uses the STOICC compiler to handle the irregular pattern of dense v.s. sparse tiles which might limit adoption in the broader community in the short term.
2. Pruned models still do not fully match the accuracy of the dense models. For example, at 45% sparsity, Llama-2 7B under PATCH reaches ~49% average accuracy compared to 54.6% in dense; Even at 25% sparsity (the least aggressive sparsity PATCH experimented with), it gets ~51.6% vs 54.6% dense. This gap is much smaller than with previous structured pruning (MaskLLM had ~6 point gap at 50%), but it’s still a performance loss.

**Questions:**

1. Have the authors considered or attempted applying PATCH to models larger than 8B, such as 13B or even 70B LLMs? If not in practice, a discussion on expected behavior would be valuable. Do the authors anticipate any new challenges at that scale?
2. Have the author evaluated integrating a low-rank adaptation during mask training (e.g., fine-tuning a small low-rank subset of weights to compensate for pruning)?
3. Could the authors compare PATCH’s learned layer-wise sparsity distribution to a heuristic allocation? Would a combined approach further benefit accuracy? Prior work (Yin et al. 2025) suggests not all layers should be pruned equally. It might strengthen the work to discuss or even experiment with a strawman baseline
4. How stable are the results across different random seeds or runs? Did the masks learned converge to similar patterns each time, or was there variance? -- Gumbel-Softmax introduces stochasticity so some note on this front would be beneficial
5. After having the final mask, have the authors considered doing a brief fine-tuning of the remaining weights to further recover accuracy?

---

> ### Author Response · Authors · 2025-11-22
>
> We thank the reviewer for their insightful feedback. In the following, we provide detailed answers to the questions and concerns raised by the reviewer.
>
> Formatting Convention: To ensure clarity, tables and figures introduced in this rebuttal are referred to simply by their number (e.g., Table 1). References to content in the original manuscript use the format `Content-Page-Number` (e.g., `Table 3-Page 7`).
>
> ---
>
> # Reliance on 2:4 Sparsity Pattern
>
> We acknowledge that our current implementation relies on the 2:4 pattern and the STOICC compiler. However, we view these as strategic choices to maximize immediate utility on current hardware rather than fundamental limitations of the PATCH framework.
>
> **Generalizability of the Framework:** PATCH is inherently pattern-agnostic. While we utilize the 2:4 pattern because it is the only semi-structured format with native Tensor Core acceleration on NVIDIA GPUs, the tile-level hybrid mechanism can easily accommodate future hardware-supported patterns (e.g., 1:4 or block-sparsity). The framework simply extends the tile distribution to $K$ classes and combines their masks via a weighted average:
> 	$$ M = M_{\text{tile}}[0] \odot \mathbf{1} + M_{\text{tile}}[1] \odot M_{1:4} + M_{\text{tile}}[2] \odot M_{2:4} + M_{\text{tile}}[3] \odot M_{3:4}$$
>
> **Adoption and STOICC:** Regarding the concern about the STOICC compiler limiting adoption, we respectfully argue that it lowers the barrier to entry. STOICC is built upon OpenAI Triton, which has become a standard, Python-based intermediate language for GPU programming in the PyTorch ecosystem. By leveraging Triton, STOICC avoids the complexity of custom CUDA kernels and ensures portability across different GPU architectures (e.g., NVIDIA and AMD). We believe this integration makes PATCH more accessible to the broader community than relying on rigid, vendor-specific libraries like cuSPARSELt.
>
> ---
>
> # Performance Gap with the Dense Model
>
> We appreciate the reviewer noting the performance gap. We view this as a reflection of the inherent trade-off between hardware-accelerated inference and model capacity, which PATCH navigates more effectively than prior methods.
>
> **Comparison to Unstructured Baselines:** While unstructured methods (e.g., Wanda, SparseGPT) often match dense accuracy, they fail to deliver wall-clock speedups on current hardware due to irregular memory access. In contrast, PATCH provides practical end-to-end acceleration (up to 1.38x) by adhering to strict hardware constraints. Among methods that actually yield speedups, PATCH achieves the closest alignment to dense accuracy reported to date.
>
> **Impact of Training constraints (PATCH-Tile vs. PATCH-Joint):** It is important to clarify that for the 7B and 8B models, we utilized the memory-efficient PATCH-Tile variant due to computational constraints. This variant freezes the fine-grained 2:4 mask and only learns tile assignments. As shown in `Table-1-Page-6` (0.5B model), PATCH-Joint (which learns both) significantly outperforms PATCH-Tile. Since larger models are typically more over-parameterized and easier to prune, we expect that training PATCH-Joint on the 7B model would significantly close the remaining gap. However, training PATCH-Joint on a 7B scale requires resources comparable to MaskLLM (~1280 A100 GPU-hours), which was beyond our available compute. We believe the results presented for PATCH-Tile represent a lower bound on the method's potential at this scale.
>
> # Training and Performance on Bigger Models
>
> We did not scale beyond 8B parameters strictly due to the high computational cost associated with training larger models end-to-end.
>
> While the 8B model could be trained on a single 80GB GPU, scaling to 70B parameters increases the computational requirements by an order of magnitude. For context, PATCH-Tile on an 8B model required approximately 96 H100 GPU-hours. Linearly extrapolating this to a 70B model (excluding the additional communication overhead of multi-node training) implies a cost of over 840+ H100 GPU-hours per run. This exceeds our available academic computational budget.
>
> However, regarding the expected behavior: **We anticipate that PATCH would perform even better at the 70B scale.** It is well-established in the literature (and observed in our own 0.5B vs. 8B comparisons) that larger models possess greater redundancy and are generally more robust to pruning. Since PATCH effectively exploits redundancy via learnable masking, applying it to a 70B model would likely yield higher accuracy retention and potentially allow for even more aggressive sparsity ratios than what is achievable on 7B/8B models.

---

> ### Author Response · Authors · 2025-11-22
>
> # Low-Rank Adaptation during Mask Training
>
> Thank you for the thoughtful suggestion. Incorporating a low-rank adaptation during mask training is indeed a promising direction. However, the additional retraining cost, including the necessary hyperparameter tuning, would be substantial, and the rebuttal period does not provide enough time to explore it thoroughly. We plan to address this in future work.
>
> ---
>
> # Sparsity Distribution Comparison with Prior Work
>
> To address this, we have conducted new experiments comparing PATCH against two state-of-the-art sparsity allocation methods: OWL [1] and AlphaPruning [2]. The resulting sparsity distributions are visualized in the newly added `Appendix-H-Page-22,23` (Figures 6, 7, and 8).
>
> To ensure a robust baseline, we performed extensive hyperparameter sweeps for both methods and selected the configurations that achieved the best perplexity when applying Wanda as the underlying pruning operator:
>
> - OWL: We swept $\lambda \in \{0.01, 0.03, 0.05, 0.08, 0.1\}$ and $M \in \{3, 5, 7, 10\}$.
> - AlphaPruning: We swept 20 values of the temperature parameter $\tau$ between 0 and 0.5.
>
> A critical distinction is that OWL and AlphaPruning are primarily designed for unstructured sparsity. While OWL includes an N:M structured variant, it is restricted to a fixed 50% global sparsity. Furthermore, OWL allocates sparsity at the block level (assigning uniform sparsity to all matrices within a Transformer block). Although the authors propose a weight-wise variant, they report, and our experiments confirm, that it yields inferior performance. In contrast, PATCH operates with tile-level granularity within a semi-structured constraint, offering a unique combination of hardware acceleration and fine-grained control.
>
> As illustrated in `Appendix-H-Page-21`, both baselines produce relatively flat sparsity distributions across the model. AlphaPruning exhibits only minor fluctuations in the middle layers, whereas PATCH discovers distinct, highly non-uniform patterns (e.g., preserving Attention layers while aggressively pruning MLP blocks).
>
> [1]  L. Yin et al., “Outlier Weighed Layerwise Sparsity (OWL): A Missing Secret Sauce for Pruning LLMs to High Sparsity,” Oct. 08, 2023, arXiv: arXiv:2310.05175. doi: 10.48550/arXiv.2310.05175.
>
> [2] H. Lu, Y. Zhou, S. Liu, Z. Wang, M. W. Mahoney, and Y. Yang, “AlphaPruning: Using Heavy-Tailed Self Regularization Theory for Improved Layer-wise Pruning of Large Language Models,” Oct. 14, 2024, arXiv: arXiv:2410.10912. doi: 10.48550/arXiv.2410.10912.
>
>
>
> ---
>
> # Performance and sparsity distribution across seeds
>
> We thank the reviewer for raising this important point regarding the stochasticity of Gumbel-Softmax. To address this, we conducted repeated runs with three different random seeds (25, 26, 42) for Qwen-2.5 0.5B and Llama-3.2 1B across all sparsity levels.
>
> As shown in Table 1 below, model performance remains consistent across seeds.. This confirms that the optimization reliably converges to minima of comparable quality.
>
>
> | Model         | Sparsity (%) | Seed 0 (current) | Seed 25 | Seed 26 | Seed 42 |
> |---------------|--------------|---------------|---------|---------|---------|
> | Qwen-2.5 0.5B | 25           | 13.47         | 13.41   | 13.38   | 13.36   |
> |               | 35           | 13.84         | 13.89   | 13.85   | 13.84   |
> |               | 45           | 14.56         | 14.59   | 14.61   | 14.49   |
> | Llama-3.2 1B  | 25           | 11.00         | 11.09   | 11.03   | 11.21   |
> |               | 35           | 11.67         | 11.72   | 11.56   | 11.86   |
> |               | 45           | 12.23         | 12.32   | 12.26   | 12.55   |
>
> Table 1: Perplexity of models using PATCH across seeds (lower is better)
>
>
>
> We have visualized the learned sparsity masks across these seeds in the newly added `Appendix-I-Page-24,25` (Figures 9, 10, and 11). The learned allocations exhibit high structural consistency:
>
> - Block-level: In all runs, the optimization consistently identifies the middle Transformer blocks as the most redundant (receiving the highest sparsity), while the initial and (especially) final blocks are pruned less.
> - Matrix-level: The allocation of sparsity between attention and MLP modules remains stable. For example, in Llama-3.2 1B (Figure 10), the Key and Value matrices consistently remain dense across all seeds, while the Query and MLP matrices absorb the majority of the sparsity.
>
> While the specific tile indices may vary slightly due to the stochastic sampling, the macroscopic pruning strategy learned by PATCH is highly reproducible and robust to initialization.

---

> ### Author Response · Authors · 2025-11-22
>
> # Fine-tuning Following Mask Training
>
> Yes, we have explored recovering accuracy via weight updates after the mask is fixed. We conducted a brief 5.6M-token run on the SlimPajama dataset so that the fine-tuning phase matches the mask search of PATCH. We have used ADAMW optimizer with learning rates chosen between (5e-4, 1e-4, 5e-5, 1e-5) and reported the best average accuracy achieved.
>
>
> As shown below in Table 2 (which we have added to `Appendix-J-Page-24`), fine-tuning consistently improves downstream task accuracy across both Qwen-2.5 0.5B and Llama-3.2 1B models.
>
> | Model | Sparsity | Method | Wiki PPL (↓) | Avg Acc (↑) |
> | :--- | :--- | :--- | :--- | :--- |
> | **Qwen-2.5 0.5B** | 45% | PATCH (Joint) | **14.56** | 40.29 |
> | | | PATCH (Joint) + FT | 14.96 | **40.87** |
> | | 35% | PATCH (Joint) | **13.84** | 41.15 |
> | | | PATCH (Joint) + FT | 14.32 | **41.59** |
> | | 25% | PATCH (Joint) | **13.47** | 42.39 |
> | | | PATCH (Joint) + FT | 13.85 | **42.55** |
> | **Llama-3.2 1B** | 45% | PATCH (Joint) | **12.23** | 42.08 |
> | | | PATCH (Joint) + FT | 12.64 | **42.94** |
> | | 35% | PATCH (Joint) | **11.67** | 42.72 |
> | | | PATCH (Joint) + FT | 12.02 | **43.50** |
> | | 25% | PATCH (Joint) | **11.00** | 43.81 |
> | | | PATCH (Joint) + FT | 11.36 | **44.61** |
> Table 2: Comparisons for Llama-3.2 1B at 45% + FT are omitted due to incomplete run data, though we expect a similar trend.
>
>
> Fine-tuning yields a consistent improvement in average zero-shot accuracy (e.g., +0.8% for Llama-3.2 1B at 25% sparsity and +0.6% for Qwen-2.5 0.5B at 45% sparsity). Interestingly, we observe a slight degradation in perplexity. We attribute this to the limited calibration data (5.6M tokens) compared to the trillions of tokens seen during pre-training: short fine-tuning can sometimes slightly drift the language modeling distribution while sharpening downstream task performance.
>
> These results confirm that PATCH creates a high-quality sparsity pattern that serves as a strong foundation. While the mask alone delivers state-of-the-art performance, subsequent fine-tuning, even with a limited budget, can further recover accuracy, offering a flexible path for users with additional compute resources.

---

### Author Response · Authors · 2025-11-22
**Thank you!**

Dear Area Chairs and Reviewers,


We sincerely thank you for the time and care you invested in reviewing our submission. We greatly appreciate the constructive feedback and the recognition of our contributions, highlighting PATCH for its (1) novel hybrid dense–sparse tile design achieving state-of-the-art accuracy–speed trade-offs for pruned LLMs [Reviewer bYHv](https://openreview.net/forum?id=RQaJrHGBAU&noteId=YNWwNoo7nz), (2) learned non-uniform sparsity allocation across layers [Reviewer Vov1](https://openreview.net/forum?id=RQaJrHGBAU&noteId=YNWwNoo7nz), and (3) clear presentation and thorough experimental analysis [Reviewer MP3s](https://openreview.net/forum?id=RQaJrHGBAU&noteId=Qu87Pu81AZ).


Your feedback helped us significantly strengthen the submission. Below is a concise summary of what we addressed:


 - `Rebuttal-A1`: Brief **fine-tuning after PATCH mask search**
 - `Rebuttal-A2`: Accuracy and throughput comparison with **FFN-only 2:4 pruning and one-shot 2:4 + fine-tuning**, showing PATCH outperforms all these baselines under similar compute.
 - `Rebuttal-A1`, `Rebuttal-A2`: Comparison with **other non-uniform sparsity allocation work**
 - `Rebuttal-A1`, `Rebuttal-A3` : Performance of PATCH **across seeds and different datasets**, showing PATCH remains consistent
 - `Rebuttal-A3` : **Throughput under smaller tile sizes**, showing why our choice of tile size is optimal

We believe these points are important for maintaining a constructive dialogue and supporting a fair assessment of our contributions. We value the reviewers’ perspectives and appreciate the thoughtful discussions their questions have sparked, which may guide future research directions.

We kindly ask that reviewers be encouraged to continue engaging in the discussion. Their insights and clarifications would significantly contribute to reaching a more complete consensus.

Thank you again for your support. We hope the remaining discussion period leads to productive exchanges, and we are available to address any additional concerns.

Best regards,

The Authors

---

### Comment · Area_Chair_X8Ka · 2025-11-25
**Please read the rebuttal and respond**

Dear reviewers,

Now that the author responses are in, could you please take a look at them and see if they address your concerns adequately?

Thank you very much.

Best,
AC

---

### Author Response · Authors · 2025-12-03
**Author Final Remarks**

**Final Remarks**:

We sincerely thank all reviewers for their thoughtful assessments and constructive suggestions, and the Area Chair for faciliting the discussion. Their feedback substantially strengthened the paper and helped clarify the contributions of PATCH.

Across the rebuttal, we addressed all major concerns with new experiments, clarifications, and revisions. We summarize again the key points below:

- **Strong baselines and fairness of comparison**. We compared PATCH with a baseline that applies one-shot 2:4 pruning only to FFN modules to test whether this simple heuristic could match PATCH. PATCH consistently outperforms FFN-only pruning at equal sparsity, and even when PATCH is applied at higher sparsity levels. Throughput is essentially identical between the two approaches, and because PATCH achieves higher accuracy at the same speed, its learned non-uniform allocation offers a strictly better accuracy–efficiency trade-off.

- **Comparison to fine-tuned one-shot baselines**. Under a fixed compute budget, PATCH decisively outperforms fine-tuned Wanda/SparseGPT 2:4 models. Learning PATCH's hybrid sparsity pattern directly has greater effect than updating weights under a fixed 2:4 mask.

- **Robustness and reproducibility**. Additional runs across multiple seeds and across SlimPajama vs. C4 demonstrate that PATCH learns stable sparsity patterns and yields consistent downstream performance.

- **Additional fine-tuning**. Short weight fine-tuning after mask learning further improves accuracy, which confirms that PATCH produces a high-quality mask. Longer fine-tuning schedules could potentially narrow the remaining gap between sparse and dense models even further.

- **Layer-wise sparsity allocation comparisons**. We added comparisons with prior non-uniform sparsity allocation methods. PATCH identifies sharper allocation patterns than existing approaches, while being the only non-uniform semi-structured sparsity allocation method that provides real end-to-end speedup.

- **PATCH's generality and practicality**. We clarified that PATCH is pattern-agnostic and not tied to 2:4 sparsity. The current choice simply aligns with available hardware support. We also highlighted that the STOICC compiler (built on Triton) makes heterogeneous tile execution accessible and easy to adopt.

- **Impact of tile size on speedup**. We clarified how tile size affects speedup. Accuracy is unchanged across tile sizes, but smaller tiles slow down GEMM kernels, so our chosen configuration provides the best performance.

We thank the reviewers and the area chair again for their valuable time. We hope these additions address all outstanding concerns.

Sincerely,

The Authors

---

### Meta-Review · Area_Chair_YRxU · 2026-01-03

**Summary:**

This paper proposes PATCH, a learnable tile-level hybrid sparsity framework that mixes dense and 2:4 sparse tiles to achieve a continuous sparsity range (0–50%), aiming to improve the accuracy–efficiency trade-off of LLM pruning while enabling real hardware speedups.

Despite strong engineering and extensive rebuttal experiments, reviewers’ concerns converged on several points:

(i) the empirical gains, while consistent, remain incremental relative to simpler baselines once matched for sparsity and throughput, raising questions about whether the added complexity is justified;

(ii) the lack of validation on larger-scale models (e.g., ≥13B/70B), although understandable from a compute perspective, leaves uncertainty about scalability and real-world relevance at the regimes where such pruning would matter most.

Based on these concerns, particularly the limited evidence that PATCH offers a clearly compelling advantage over simpler or more established alternatives at scale, I recommend reject.

**Reviewer Concerns:**

Reviewer bYHv: Seed robustness, comparison to heuristic layer-wise allocation, feasibility of post-mask fine-tuning, and discussion of scaling beyond 8B, and the reliance on 2:4 + STOICC are discussed. The core limitation remains that results still trail dense models and the method’s practical value is tied to specialized 2:4 support/compilers; the “expected to work better at 70B” argument is plausible but not empirically validated.

Reviewer Vov1: The main technical baseline concerns were largely met: they implemented FFN-only 2:4 pruning baselines, added matched-throughput comparisons under the same backend, compared against fine-tuned one-shot baselines under a fixed compute budget, and clarified 3:4 vs. PATCH-25%. This was convincing enough that the reviewer increased the score (2→6). However, generative evaluation (e.g., EvalPlus) is acknowledged but not actually provided during rebuttal; Hopper/H100 benchmarking is still missing due to backend limitations; these leave some uncertainty about real-world utility on current hardware/software stacks.

Reviewer MP3s: Concrete compute-cost reporting for mask learning, tile-size impact on speed, dataset robustness, and why 80–90% sparsity is structurally incompatible with the 2:4 hybrid design (0–50% bound) are discussed. But the reviewer’s biggest concern remains, i.e., lack of evidence on larger models. The rebuttal provides a reasonable compute argument and cites precedents, but it does not replace direct empirical validation or a stronger scaling study, so uncertainty about scalability/practical impact at the most relevant deployment regimes persists.

**Reviewer Scores:**

Reviewer Vov1: 2 --> 6

Others remain unchanged.

---

### Decision · Program_Chairs · 2026-01-26

Reject